# The CHK1 inhibitor prexasertib in *BRCA* wild-type platinum-resistant recurrent high-grade serous ovarian carcinoma: a phase 2 trial

Elena Giudice [1,2,9], Tzu-Ting Huang [1,9], Jayakumar R. Nair[1,9], Grant Zurcher[1], Ann McCoy[1], Darryl Nousome[3], Marc R. Radke[4], Elizabeth M. Swisher [4], Stanley Lipkowitz[1], Kristen Ibanez [1], Duncan Donohue[5], Tyler Malys[5], Min-Jung Lee[6], Bernadette Redd [7], Elliot Levy[8], Shraddha Rastogi[6], Nahoko Sato [6], Jane B. Trepel[6] & Jung-Min Lee [1] ✉

The multi-cohort phase 2 trial NCT02203513 was designed to evaluate the clinical activity of the CHK1 inhibitor (CHK1i) prexasertib in patients with breast or ovarian cancer. Here we report the activity of CHK1i in platinum-resistant high-grade serous ovarian carcinoma (HGSOC) with measurable and biopsiable disease (cohort 5), or without biopsiable disease (cohort 6). The primary endpoint was objective response rate (ORR). Secondary outcomes were safety and progression-free survival (PFS). 49 heavily pretreated patients were enrolled (24 in cohort 5, 25 in cohort 6). Among the 39 RECISTv1.1-evaluable patients, ORR was 33.3% in cohort 5 and 28.6% in cohort 6. Primary endpoint was not evaluable due to early stop of the trial. The median PFS was 4 months in cohort 5 and 6 months in cohort 6. Toxicity was manageable. Translational research was an exploratory endpoint. Potential biomarkers were investigated using pre-treatment fresh biopsies and serial blood samples. Transcriptomic analysis revealed high levels of DNA replication-related genes (*POLA1*, *POLE*, *GINS3*) associated with lack of clinical benefit [defined post-hoc as PFS < 6 months]. Subsequent preclinical experiments demonstrated significant cytotoxicity of *POLA1* silencing in combination with CHK1i in platinum-resistant HGSOC cell line models. Therefore, *POLA1* expression may be predictive for CHK1i resistance, and the concurrent POLA1 inhibition may improve the efficacy of CHK1i monotherapy in this hard-to-treat population, deserving further investigation.

Ovarian carcinoma is the most lethal gynecologic malignancy in developed countries, with high-grade serous ovarian carcinoma (HGSOC) being the most common subtype[1]. Despite optimal debulking surgery and platinum-based chemotherapy, approximately 80% of HGSOC patients relapse after first-line treatment[2]. Platinum-resistant recurrent HGSOC, particularly in *BRCA* wild-type (BRCAwt) cases (~75% of all HGSOCs[3,4]), poses the greatest challenge with limited treatment

options[5], highlighting an unmet need to develop novel therapeutic agents.

One of the molecular characteristics of platinum-resistant HGSOC is increased replication stress (RS)[6]. RS can be caused by any events disrupting DNA replication but is traditionally defined as the slowing or stalling of replication-fork progression and/or DNA replication[7–9]. HGSOC cells have changes in many genes involved in DNA replication,

e.g., universal *TP53* mutations with consequently defective G1/S cell-cycle regulation[3], *CCNE1* amplification, and loss of *RB1* and *NF1*[10]. As a result, HGSOC cells highly depend on other S- and G2/M checkpoint pathways, e.g., Ataxia Telangiectasia and Rad3-related kinase (ATR)/ checkpoint kinase 1 (CHK1) signaling for DNA replication[11]. Accordingly, upregulation of the ATR/CHK1 pathway is observed in drug-resistant HGSOC[12,13], underscoring the therapeutic potential of ATR/ CHK1 pathway inhibitors in platinum-resistant disease.

In HGSOC, *CCNE1* copy number (CN) gain/amplification or over-expression has been studied as prognostic or predictive biomarkers[14–17]. Kang et al. reported that high *CCNE1* amplification (>8 CN) was associated with shorter survival[16]. *CCNE1* amplified/over-expressed tumors also have been shown to be sensitive to WEE1 inhibitors[18,19]. However, the clinical utility of *CCNE1* amplification/ overexpression as a predictive biomarker requires more investigation as some data do not support its predictive value for ATR/CHK1 inhibitors[15,20–22]. Those conflicting observations suggest that a single-biomarker selection to target a specific population is challenging because RS is a dynamic process and there is no clear consensus about RS-related biomarkers for predicting response to ATR/CHK1 blockade. Another effort includes investigating gene signatures potentially reflective of high-RS[10,23]. The phase 2/3 clinical trial of gemcitabine alone versus gemcitabine and ATR inhibitor (ATRi) berzosertib combination in HGSOC suggested a high-RS molecular signature (*MYC*, *MYCL1*, *ERBB2* and *KRAS* amplifications, *RB1* or *CDKN2A* two-copy loss, or *NF1* mutations) may predict gemcitabine response in patients with a high-RS background, while those with a low-RS may benefit from a combination strategy[10]. Collectively, those data suggest that additional efforts are needed to define RS and its clinical utility as a predictive biomarker.

Among many genes involved in DNA replication, key players include DNA polymerases alpha (POLA), delta (POLD), and epsilon (POLE), which are the three minimally required DNA polymerases essential to complete the DNA synthesis, belonging to the B-family, and characterized by distinct synthetic capacities. POLE and POLD are the main DNA replicases while POLA is a DNA-primase involved in DNA synthesis initiation. Initiation of DNA replication is a critical step, which requires the unwinding of DNA strands by the Cdc45-MCM-GINS (CMG) helicase complex to allow the bidirectional DNA replication, and the POLA-primase complex to generate primers for the DNA synthesis, successively extended by DNA replicases in a continuous (POLE, leading strand) or discontinuous (POLD, lagging strand) DNA synthesis process[24,25]. Those error-free, B-family polymerases concert with the CMG complex to transmit the genomic information through a high-fidelity replication process and a proofreading activity[26,27] to avoid RS and mitotic catastrophe[28]. Therefore, the clinical applicability of DNA polymerases as a biomarker or as a therapeutic target in the HGSOC setting requires further investigation.

We previously reported the clinical activity of the CHK1i pre-xasertib in heavily pretreated BRCAwt HGSOC patients[29] including both platinum-sensitive and -resistant diseases from the BRCAwt cohort of the phase 2 NCI single-center basket trial (NCT02203513). Based on the early activity signal of CHK1i in BRCAwt HGSOC, we opened the new BRCAwt cohorts focused on platinum-resistant HGSOC in this NCI basket trial to confirm the activity of CHK1i as well as to conduct more biomarker analyses beyond *CCNE1* alterations.

In this study, we report the results from the new BRCAwt platinum-resistant HGSOC cohorts and provide translational studies that may contribute to further tailoring treatment, with the identification of genomic and transcriptomic features correlating with CHK1i response and resistance. We also explore our hypothesis that *POLA1* inhibition may induce additive or synergistic lethality with CHK1i in platinum-resistant HGSOC preclinical models based on the transcriptomic data from fresh tissue samples. Our results indicate that *POLA1* expression may be associated with CHK1i resistance, and the

concurrent *POLA1* inhibition may improve the efficacy of CHK1i in this hard-to-treat population, requiring further investigation.

## Results

### Patient demographics and treatment
Between January 25, 2017 and March 23, 2020, 49 BRCAwt platinum-resistant recurrent HGSOC patients were enrolled, including 24 patients with biopsiable disease (cohort 5) and 25 patients without safely biopsiable disease (cohort 6). Those in cohort 5 underwent pre-treatment core biopsies (Fig. 1a), while blood samples were collected pre- and on-treatment from all in cohorts 5 and 6 (Fig. 1b). The trial was stopped early due to COVID-19 and termination of investigational drug supplies by the company before enrolling the planned numbers. Therefore, the participants from cohorts 5 and 6 are also reported together as a combined dataset to have a reasonable number for data analysis given that the clinical results were sufficiently similar between the two cohorts (Supplementary Data 1). Baseline demographic and disease characteristics are described in Supplementary Table 1. The absence of any somatic/germline BRCA1/2 mutations was confirmed by a CLIA-certified laboratory (multi-gene panels or individual testing). Notably, all patients were heavily pretreated, with a median of 4 prior systemic therapies (IQR 3–7).

### Antitumor activity and safety
All 49 patients received at least one dose of prexasertib. Ten patients were not assessable for tumor response per Response Evaluation Criteria In Solid Tumors (RECIST) 1.1 criteria because of no restaging CT scans after 2 cycles of treatment due to withdrawal of consent or intercurrent illness during cycle 1 (Fig. 2a, b). Among the RECIST-evaluable patients ($n = 39$), the objective response rate (ORR) was 30.8% (12/39, 95% confidence interval [CI]: 17–47.6), with 33.3% in cohort 5 (6/18) and 28.6% in cohort 6 (6/21), respectively. Disease control rate (DCR), defined by the sum of patients with partial response (PR) and stable disease (SD) ≥ 6 months, was 56.4% (22/39), with 44.4% (8/18) in cohort 5 and 66.7% (14/21) in cohort 6. The median progression-free survival (PFS) was 5 months, with 4 months in cohort 5 ($n = 18$) and 6 months in cohort 6 ($n = 21$) (Fig. 2c, d and Supplementary Data 1). In the intention-to-treat (ITT) population ($n = 49$), an ORR of 24.5% (12/49) and a DCR of 44.9% (22/49) were observed. These results were similar to our previous report in which ORR was 31.6% (6/ 19) in the platinum-resistant HGSOC, and 28.6% (8/28) in the ITT population[29], confirming the therapeutic potential of CHK1i in this population. Of 12 who had PR per RECIST criteria, the median duration of response (DoR) was 5 months (95% CI: 3–11), with 6.5 months in cohort 5 ($n = 6$) and 4.5 months in cohort 6 ($n = 6$), respectively.

Any grade treatment-related adverse events (TRAEs) were listed in Supplementary Data 2. The most common (in >10% of patients) grade 3 or 4 TRAEs were hematological toxicities, such as neutropenia (42/49, 85.7%), leukocytopenia (38/49, 77.6%), lymphocytopenia (23/49, 46.9%), thrombocytopenia (20/49, 40.8%), anemia (15/49, 30.6%) and febrile neutropenia (6/49, 12.2%), consistent with previous reports[29]. Of note, granulocyte colony-stimulating factors were given after checking the nadir on cycle 1 day 8 to avoid treatment delay or dose reduction.

### Genomic profiling does not reveal an association between gene alterations and CHK1i response
While we confirmed the clinical efficacy of CHK1i in platinum-resistant BRCAwt HGSOC, we found no significant correlation between CHK1i response and *CCNE1* amplifications ( > 8 CN) or mRNA overexpression (Fig. 2e and Supplementary Data 3-5). Only two cases were found *CCNE1* amplified with one (11 CN) achieving clinical benefit (CB, defined post hoc as PFS ≥ 6 months) and another (9 CN) with no clinical benefit (NCB, PFS < 6 months). We chose six months of PFS as a cut-off for CB because single-agent chemotherapy yields 3–4 months of PFS in a heavily pretreated platinum-resistant HGSOC patient population[30,31].

To investigate other gene alterations in DNA damage repair (DDR) and other survival pathways, we sequenced pre-treatment fresh biopsies with the BROCA-GOv1 gene panel (Fig. 2e and Supplementary Data 3). Patients with CB were also characterized by *MYC* CN gain (*n* = 1, 6 CN) and *CDK12* mutation (*n* = 1); however, those gene alterations were observed in patients without CB as well (*MYC*: *n* = 5, 5–8 CN; *CDK12*: *n* = 1) (Fig. 2e and Supplementary Data 4). In contrast with previously reported RS signatures predictive of DDR inhibitors' response[10,32], *KRAS* amplification (*n* = 2), *NF1* copy loss (*n* = 1), *RAD51* mutations (*n* = 1), were found in patients without CB (Fig. 2e and Supplementary Data 4).

In addition, we performed whole exome sequencing (WES) to further study the genomic profiles for possible correlations with CB in DDR-related genes beyond the already known predicted pathogenetic/pathogenetic variants detected by BROCA-GOv1 assay (Supplementary Table 2, and Supplementary Fig. 1). Variants of uncertain significance (VUS) were found in *RB1* and *BRIP1* genes (missense mutations) in two different patients without CB (Supplementary Fig. 1). Moreover, among the 76 genes not included in the BROCA-GOv1 panel, *RECQL4* VUS (missense mutation) was found in one achieving CB, while a pathogenic *PIK3CA* missense mutation was observed in one patient without CB although these are exploratory analyses (Supplementary Fig. 1). Additionally, no association with CB was detected among all the 126 gene mutations analyzed (Supplementary Table 2).

### Transcriptomic profiles exhibit the association of high-fidelity DNA replication machinery with CHK1i resistance

Next, we conducted transcriptomic analysis on pre-treatment fresh biopsies through RNA sequencing (RNAseq, Supplementary Data 5) to identify signaling pathways that might correlate with CHK1i resistance

or response. Specifically, the DNA replication pathway (rank 2, false-discovery rate [FDR] q = 0.025, | normalized enrichment score (NES)| = 1.81, and nominal *P* = 0.001) was significantly enriched in NCB (Fig. 3a and Supplementary Data 6) using the Gene Set Enrichment Analysis (GSEA) with Kyoto Encyclopedia of Genes and Genomes (KEGG) database[33]. Among others, *POLE* (enrichment score [ES] = 0.10) and *POLA1* (ES = 0.21) contributed to the core enrichment of the DNA replication pathway (Fig. 3b). *POLE* (*P* = 0.011), *POLA1* (*P* = 0.037) along with *GINS3* (*P* = 0.02) which belong to the CMG helicase complex, directly binding to POLA and POLE as part of the replisome, to restart DNA replication upon stalling DNA synthesis by RS[34–36], were upregulated in NCB (Fig. 3c and Supplementary Data 7). We also investigated the clinical relevance of *POLE* and *POLA1* in HGSOC by using public datasets. In HGSOC, 7% exhibited elevated levels of *POLE* or *POLA1* mRNA (Supplementary Fig. 2a). Elevated *POLE* mRNA levels correlated with worse PFS in platinum-treated HGSOC patients (median 15.01 months vs. 18.23 months; log-rank *P* = 0.012, HR = 1.36 [1.07–1.74], Supplementary Fig. 2b), suggesting a potential association of *POLE* expression with clinical outcomes in HGSOC. Accordingly, it has been reported that lower protein expression of B-family DNA polymerases is associated with CHK1i sensitivity. Concomitant inhibition of CHK1 and B-family polymerases induces RS and cell death in lung and colorectal cancer preclinical models[22]. Those exploratory results led us to hypothesize that tumor cells resistant to CHK1i might induce high tolerance to RS by upregulating the high-fidelity replication machinery. Hence, POLA1 and POLE are critical for the completion of DNA synthesis in BRCAwt platinum-resistant HGSOC. To test this hypothesis, we evaluated the biological function of these genes in cell line models for a proof-of-concept because the small sample size and

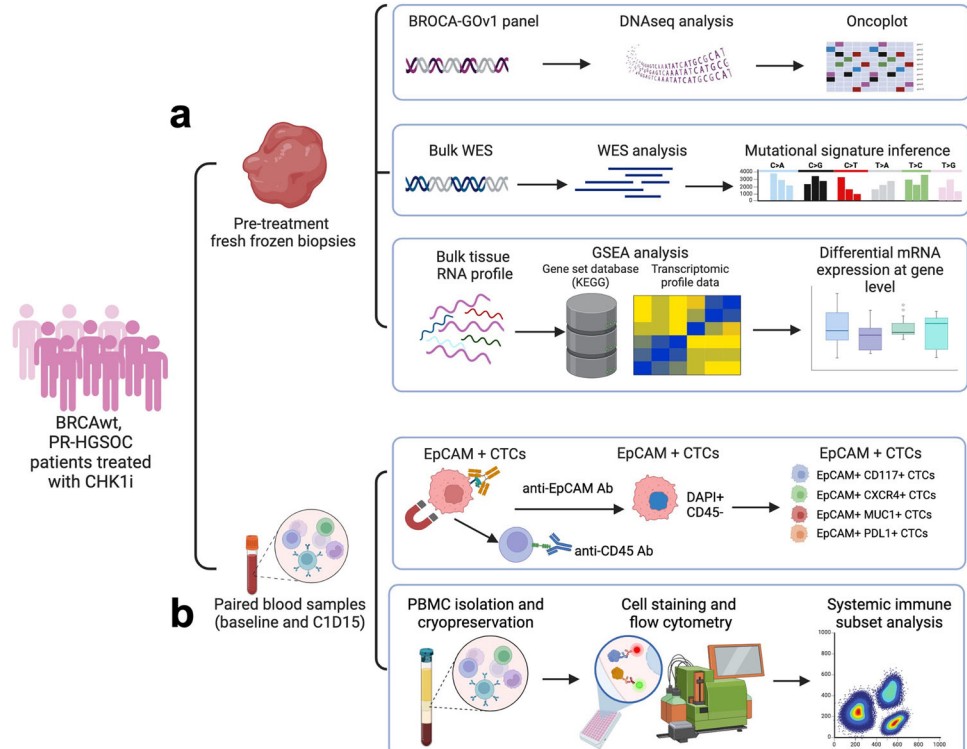

**Fig. 1 | Correlative study endpoints analyses workflow. a** Workflow to detect molecular correlates to CHK1i treatment. Pre-treatment fresh core biopsies were collected from patients with safely biopsiable diseases enrolled in the clinical trial (NCT02203513). Genomic (BROCA GOv.1 panel and WES) and transcriptomic analyses were performed to identify the molecular characteristics between patients with clinical benefit (PFS ≥ 6 months) and no clinical benefit (PFS < 6 months). **b** Workflow to detect pharmacodynamic biomarkers reflecting CHK1i therapy. Paired blood samples were collected at baseline and at C1D15 to evaluate dynamic changes in EpCAM+ CTCs and immune cell subsets in patients with and without clinical benefit. The figure was created with BioRender.com. Abbreviations: BRCAwt *BRCA* wild-type, C1D15 Cycle 1 Day 15, CHK1i CHK1 inhibitor, CTCs circulating tumor cells, DNAseq DNA sequencing, EpCAM epithelial cell adhesion molecules, GSEA gene set enrichment analysis, PBMC peripheral blood mononuclear cells, PR-HGSOC platinum-resistant high-grade serous ovarian cancer, WES whole exome sequencing.

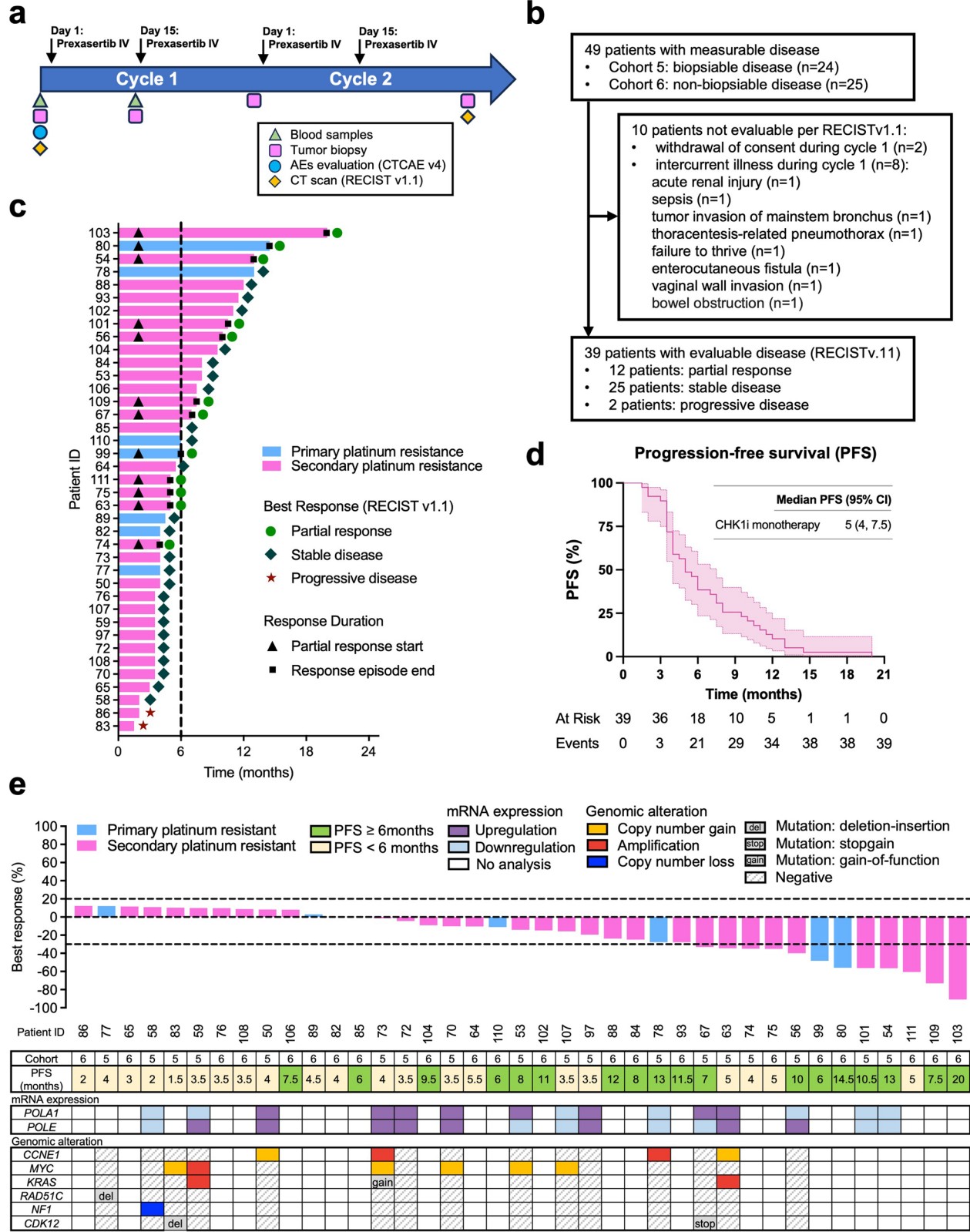

the paucity of biopsy samples make an FDR correction and the protein level validation hard to estimate (Supplementary Data 7).

**Targeting B-family polymerases induces significant cytotoxicity with CHK1i in BRCAwt, platinum-resistant HGSOC cells**

First, we treated BRCAwt platinum-resistant HGSOC cell lines (OVCAR3 and OVCAR5)[37,38] with aphidicolin, a pan-inhibitor of B-family polymerases with minimal toxicity in ovarian cancer cells up to 25 μM[39,40].

Aphidicolin monotherapy (15 μM) induced modest growth inhibition (-30%), and this effect was not further enhanced by a high concentration (30 μM; Fig. 4a). Notably, the combination of aphidicolin and CHK1i significantly inhibited cell growth in both cell lines ($P < 0.001$, Fig. 4a). Sublethal concentration of CHK1i (0.8 nM) combined with a low-concentration aphidicolin (15 μM) markedly inhibited growth by 75% in OVCAR3 and 61% in OVCAR5, compared to aphidicolin alone (29% and 25%, respectively, $P < 0.001$) and this cytotoxicity was CHK1i

**Fig. 2 | Clinical trial design and antitumor activity. a** Clinical trial with correlative study endpoints: prior to prexasertib administration, blood and tumor samples were collected for correlative study endpoints, and blood samples were further obtained at C1D15. AEs (CTCAE v4) were evaluated at each study drug administration at cycle 1, and every 4 weeks for subsequent cycles. CT scans were performed every 2 cycles (RECIST v1.1 evaluation). **b** The CONSORT flow diagram. Overall, 49 patients were enrolled in the study including 24 patients in the biopsy cohort 5 and 25 patients in the non-biopsy cohort 6. 39 patients were evaluable for tumor response per RECIST v1.1 criteria. **c** Duration of treatment: swimmer plot showing the duration of treatment (time in months) with prexasertib monotherapy for each individual RECIST-evaluable patient ($n = 39$). **d** PFS of each patient is shown. PFS was estimated using the Kaplan–Meier method with 95% CI. **e** Waterfall plot showing the best responses in 39 RECIST-evaluable patients. The horizontal

dotted line indicates the threshold for partial response (30% reduction in tumor size from baseline). Sequencing was conducted and analyzed on fresh pre-treatment tissue samples with optimal quality. *POLA1* and *POLE* mRNA levels were measured by RNAseq (Supplementary Data 5). Upregulation is defined as expression ≥median, and downregulation as expression <median for each gene (cohort 5, $n = 15$). DNAseq exhibited the genetic alteration of genes related to DDR (cohort 5, $n = 15$). Abbreviations: AEs adverse events, CTCAE Common Terminology Criteria for Adverse Events, *CCNE1* cyclin E1, *CDK12* cyclin-dependent kinase 12, CI confidence interval, DDR DNA damage repair, DNAseq DNA sequencing, IV intravenously, *KRAS* Kirsten rat sarcoma virus, *MYC* Myc proto-oncogene, *NF1* neurofibromatosis type 1, PFS progression-free survival, *POLA1* DNA polymerase alpha 1, *POLE* DNA polymerase epsilon, *RAD51C* RAD51 paralog C, RECIST Response evaluation criteria in solid tumors, RNAseq RNA sequencing.

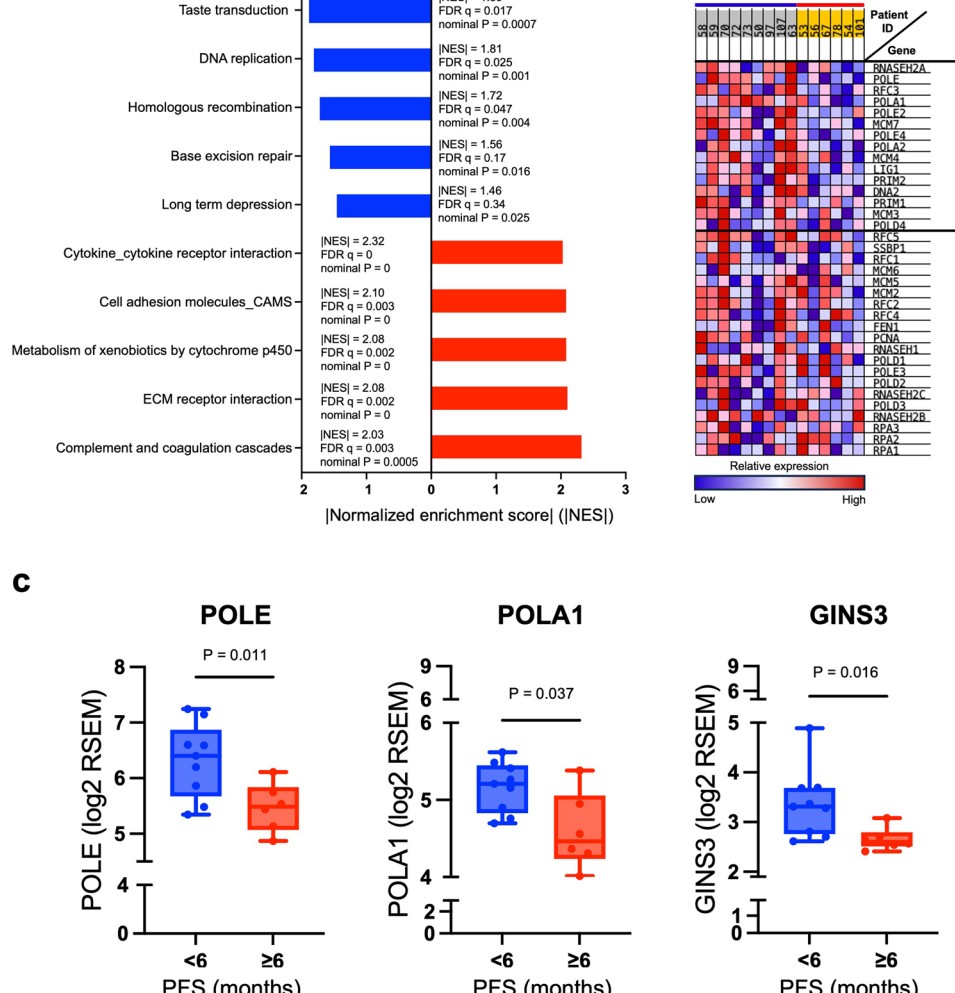

**Fig. 3 | DNA replication machinery is associated with CHK1i resistance.**
**a** Analysis of bulk RNAseq indicated no individual gene differentially expressed between CB (PFS ≥ 6 months, $n = 6$) and NCB (PFS < 6 months, $n = 9$) after adjustment of multiple testing (FDR q = 0.87–1, Supplementary Data 7), possibly due to the small sample size. We also conducted GSEA of RNAseq data to identify pathways that might contribute to CHK1i resistance and response. The bar plot shows the top five KEGG gene sets associated with CB ($n = 6$) or NCB ($n = 9$). NES scores, nominal *p*-values, and FDR *q*-values shown in the figure were calculated by GSEA software (Supplementary Data 6). **b** The DNA replication pathway was enriched in patients with NCB ($n = 9$) versus CB ($n = 6$). Genes in the core enrichment of the DNA replication pathway were shown (right). **c** The mRNA levels of genes were

analyzed from RNAseq in patients with CB ($n = 6$) versus NCB ($n = 9$). High mRNA expression of DNA replication pathway-related genes, *POLA1*, *POLE*, *GINS3* are significantly associated with no clinical benefit. A regular *t*-test (two-sided) was used for the raw *P*-value (Supplementary Data 7). The boxes extend from min to max values, with the median depicted by a horizontal line. Source data are provided as a Source Data file. Abbreviations: CB clinical benefit, CHK1i CHK1 inhibitor, FDR false-discovery rate, GSEA gene set enrichment analysis, *GINS3* GINS Complex Subunit 3, *MCM7* minichromosome maintenance complex component 7, NCB no clinical benefit, NES normalized enrichment score, RNAseq RNA sequencing, RSEM RNAseq by expectation-maximization, PFS progression-free survival, *POLA1* DNA polymerase alpha 1, *POLE* DNA polymerase epsilon.

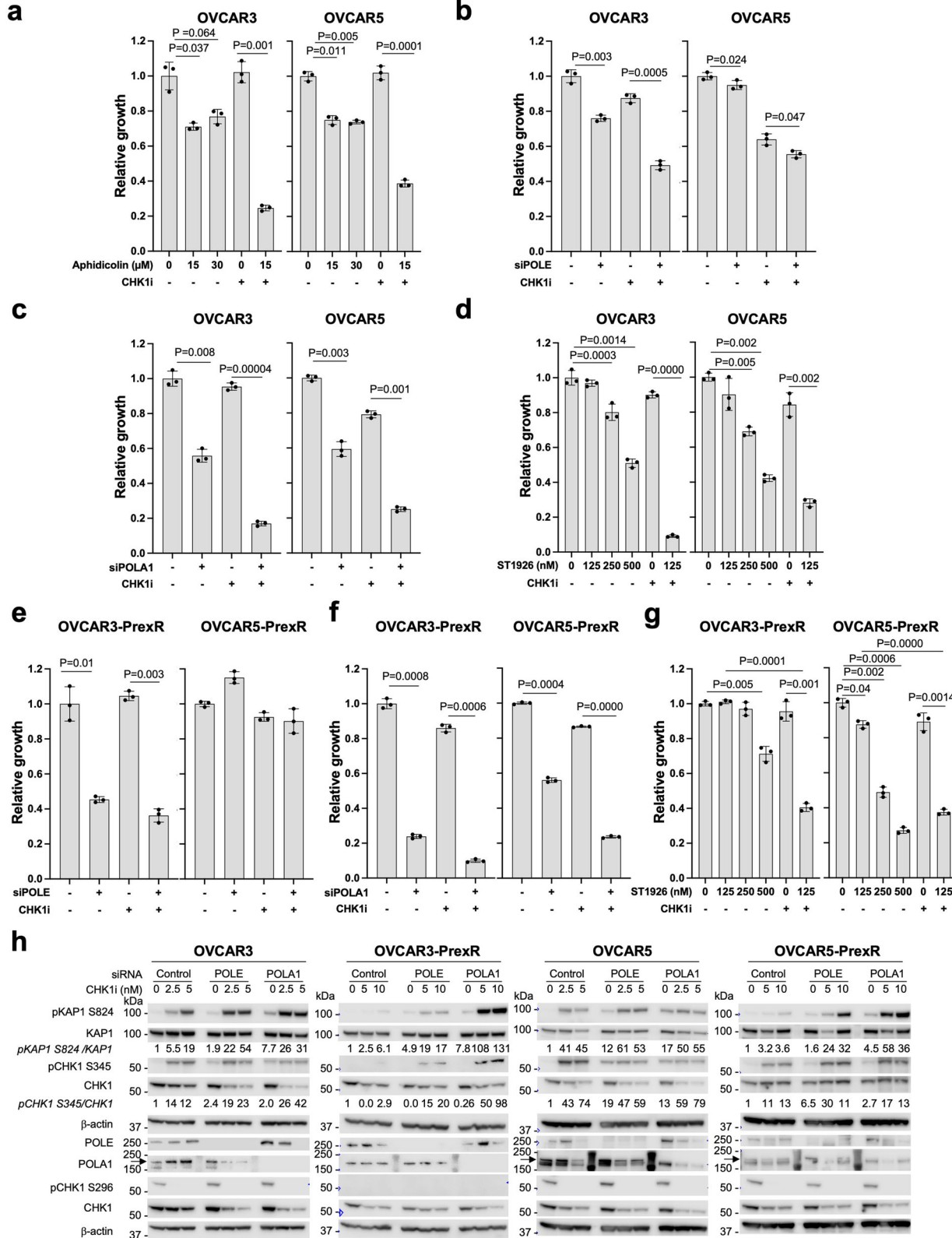

concentration-dependent (Supplementary Fig. 3a). These findings suggest that B-family polymerase activity may involve CHK1i sensitivity.

### *POLA1* expression inversely correlates with CHK1i sensitivity in BRCAwt platinum-resistant HGSOC cell lines

To identify which DNA polymerase contributed most to increased CHK1i sensitivity, we used specific small-interfering RNA (siRNA) pools

because of the differing roles of POLA1 and POLE in DNA replication. POLE is essential for later replication stages, with its depletion having minimal impact on CMG assembly and DNA initiation[41]. POLA1 is critical for coordinating DNA strand unwinding and lagging strand repair, and its inhibition can lead to deleterious events such as single-strand DNA (ssDNA) accumulation[42]. In our models, *POLE* silencing showed either low (5%, *P* = 0.024, OVCAR5) or modest (24%, *P* = 0.003,

**Fig. 4 | Targeting B-family polymerases induces significant cytotoxicity with CHK1i in BRCAwt platinum-resistant HGSOC cells. a** XTT assays were performed with a gradient of aphidicolin with or without CHK1i prexasertib (0.8 nM) and survival plotted relative to untreated ($n = 3$). **b, c** XTT assays were performed with siRNAs that target either *POLE* (**b**, $n = 3$) or *POLA1* (**c**, $n = 3$). **d** XTT assays were done similarly with increasing concentrations of POLA1 specific inhibitor ST1926 (0–500 nM) and survival plotted with ST1926 alone or in combination with sub-lethal concentration of CHK1i (1.5 nM) ($n = 3$). **e−g** Relative inhibition of PrexR cell lines with siRNA specific to *POLE* (**e**, $n = 3$), to *POLA1* (**f**, $n = 3$), or to ST1926 (**g**, n = 3) with or without CHK1i (6.25 nM). **h** Western blot analysis of protein extracts from cells treated with either control or those treated with specific siRNAs targeting either *POLE* or *POLA1* for 48 h prior to be treated with CHK1i overnight at the concentrations mentioned ($n = 3$). Densitometric analysis was performed using ImageStudio software. Phosphorylated proteins were quantified relative to total proteins, normalized to β-actin, and then expressed relative to loading control β-actin. Data from **a−g** were analyzed using a standard Student's *t*-test (two-sided). Data are shown as mean ± SD. Source data are provided as a Source Data file. Abbreviations: CHK1i CHK1 inhibitor, DDR DNA damage repair, HGSOC high-grade serous ovarian cancer, PrexR prexasertib-resistant, *POLA1* DNA polymerase alpha 1, *POLE* DNA polymerase epsilon.

---

OVCAR3) growth inhibition relative to control siRNAs (Fig. 4b). However, combining *POLE* silencing with CHK1i induced ~50% growth inhibition in OVCAR3 cells (Fig. 4b), that augmented further with increasing CHK1i concentrations (Supplementary Fig. 3b).

Interestingly, both OVCAR3 and OVCAR5 exhibited significant growth inhibition upon *POLA1* silencing alone (40–45%, $P < 0.01$) and was further enhanced when combined with sublethal CHK1i concentrations (83% and 75%, respectively, $P < 0.001$) (Fig. 4c and Supplementary Fig. 3c). Moreover, the specific POLA1 inhibitor ST1926 (125–500 nM), a next-generation adamantyl retinoid[43,44], yielded concentration-dependent growth inhibition (Fig. 4d and Supplementary Fig. 3d). CHK1i sensitivity was significantly enhanced across all ST1926 concentrations ($P < 0.001$–0.01, Supplementary Fig. 3d). Together, our data underscore the therapeutic potential of targeting POLA1 to improve CHK1i sensitivity in BRCAwt platinum-resistant HGSOC cells.

## Targeting B-family polymerases reverses prexasertib resistance in BRCAwt platinum-resistant HGSOC cell lines

To determine if targeting B-family polymerases can overcome CHK1i resistance, we used CHK1i-resistant OVCAR3 (OVCAR3-PrexR) and OVCAR5 (OVCAR5-PrexR), developed through gradual prexasertib exposure[45]. *POLE* silencing inhibited growth in both OVCAR3 and OVCAR3-PrexR ($P < 0.01$), with limited impact on OVCAR5-PrexR (Fig. 4e and Supplementary Fig. 4a). Also, silencing *POLE*-sensitized OVCAR3-PrexR cells to CHK1i in a concentration-dependent manner, while it did little to reverse CHK1i resistance in OVCAR5-PrexR (Supplementary Fig. 4a).

Notably, *POLA1* silencing significantly inhibited OVCAR3-PrexR (74%, $P = 0.0008$) and OVCAR5-PrexR (44%, $P = 0.0004$) growth (Fig. 4f). Although CHK1i alone did not induce PrexR cell death, combining it with *POLA1* silencing led to substantial growth inhibition (Fig. 4f and Supplementary Fig. 4b), highlighting POLA1's key role in CHK1i resistance.

## POLA1 inhibition to overcome CHK1i resistance in BRCAwt platinum-resistant HGSOC cell lines

Next, we assessed the pharmacologic inhibition of POLA1 and CHK1 in PrexR cells to exploit the therapeutic potential of POLA1 inhibitors. ST1926 alone induced concentration-dependent growth inhibition in both PrexR cell lines (Fig. 4g and Supplementary Fig. 4c). Even at the lowest concentration of ST1926 (125 nM), its combination with CHK1i (6.5 nM) significantly reduced OVCAR3-PrexR and OVCAR5-PrexR growth (60% and 50%, respectively, $P < 0.001$) (Fig. 4g and Supplementary Fig. 4c), aligning with *POLA1* silencing outcomes, suggesting its pivotal role in CHK1i resistance.

In immunoblot analysis, CHK1i alone triggered compensatory ATM and ATR activation (elevated levels of pKAP1-S824 and pCHK1 S345, respectively) in OVCAR5-PrexR while OVCAR3-PrexR exhibited ATM activation with subdued ATR activity (unchanged pCHK1 S345) (Fig. 4h). This may be partly due to limited ATR activity contributing to acquired CHK1i resistance in OVCAR3-PrexR, warranting further investigation. Additionally, silencing either polymerase resulted in increased pKAP1 and pCHK1 levels which were further augmented upon CHK1i treatment across all cell lines (Fig. 4h). Collectively, these results support further investigation of POLA1 inhibition as a therapeutic strategy to overcome CHK1i resistance.

## Mutational signature analysis and transcriptomic profiles revealed the association of error-prone translesion synthesis (TLS) with clinical benefit

Considering the involvement of the high-fidelity DNA replication machinery in CHK1i resistance, we hypothesized that the DNA damage tolerance (DDT) pathways also may involve CHK1i response as the DDT pathways promote the bypass of ssDNA lesions to rescue stalled replication forks[9]. Among those DDT pathways, the error-free template switching is activated during the physiological cell-cycle, while the error-prone TLS system is necessary under RS in case of large amounts of DNA damage[46]. For this, we evaluated the mutational signatures[47] by the Catalogue of Somatic Mutation in Cancer (COSMIC) Mutational Signature[48] (https://cancer.sanger.ac.uk/signatures/), which consists of four variant classes (single base substitutions (SBS), doublet base substitutions (DBS), small insertions and deletions (ID), and CN). In pre-treatment biopsy samples, we found significantly higher SBS7d and ID13 mutational signatures in the CB group compared to the NCB group (Supplementary Fig. 5a and Supplementary Table 3). Those two mutational signatures are closely linked with ultraviolet (UV) light exposure[48] (Supplementary Table 3), with the former being more likely a consequence of the TLS pathway[48]. These data support our notion that patients benefitting from CHK1i might depend on TLS, as ATR/CHK1 signaling is essential for post-TLS DNA repair in cells.

Given the association between the TLS-associated mutational signature and CHK1i response, we next questioned whether mRNA expression of TLS-related genes correlated with CHK1i response. Although transcriptome profiles did not exhibit any significant differences in the expression of TLS polymerase genes, we observed a higher mRNA expression of the Pol δ-interacting protein 2 (POLDIP2) gene in patients with CB. PolDIP2 protein activates TLS DNA polymerases and regulates the relative usage of error-prone TLS over error-free template switching by HR[49]. Furthermore, we observed mRNA over-expression of the DNA damage inducible *GADD45B* gene, activated by environmental stresses, including UV, and regulates p38/JNK mediated apoptosis[50]. Finally, a higher expression of *CASPASE3* was seen in the CB group, suggesting active stress-induced apoptosis signaling (Supplementary Fig. 5b and Supplementary Data 7). Those preliminary observations suggest that bulky DNA-damaged cells with relatively higher use of TLS polymerases than high-fidelity DNA polymerases might activate stress-induced genes. Those findings led us to hypothesize that the error-prone TLS pathway might contribute to CHK1i response in part, thus warranting investigation.

## Circulating tumor cells (CTCs) and immune cells subsets

Lastly, we evaluated the dynamic changes of CTCs and systemic immune cells upon CHK1i treatment. Blood samples were collected at baseline, and at 6–24 h after the second administration of prexasertib (cycle 1 day 15 [C1D15]). Specifically, CTCs of epithelial ovarian cancer were analyzed, marked as epithelial cell adhesion molecules (EpCAM+) CTCs and MUC1, and as other surface markers indicating epithelial-

mesenchymal transition (CD117, CXCR4)[51–53], or tumor immune escape (PDL1)[54]. Of note, improved median PFS was found in patients with decreased EpCAM+ MUC1+ CTCs compared to those with no change or increase in EpCAM+ MUC1+ CTCs from baseline to C1D15 (median PFS: 7.5 versus 4 months; $P = 0.02$, hazard ratio: 0.41, 95% CI 0.20-0.86, Supplementary Fig. 6a). Overall CTC numbers did not significantly change on C1D15 in both CB and NCB groups (Supplementary Fig. 6b).

Also, the NCB group exhibited a significant increase of monocytic myeloid-derived suppressor cells (M-MDSCs, $P < 0.01$) and classical monocytes (CM) ($P < 0.001$) from baseline to C1D15 compared to the CB group (Supplementary Fig. 7a, b). At C1D15, a higher percentage of M-MDSCs and CM were observed in patients with NCB compared to those with CB ($P < 0.01$ and $P < 0.01$, respectively) (Supplementary Fig. 7a, b).

Regarding T-cell modulation, we observed a decrease of the activated proliferating (HLA-DR+Ki67+, PD1+Ki67+, and ICOS+Ki67+) CD4+ or CD8+T-cell population from baseline to C1D15 ($P < 0.05$) in NCB group (Supplementary Fig. 8a, b). Also, the expression of the suppressive functional marker TIM-3 on CD8+ Tregs decreased from baseline to C1D15 ($P < 0.05$) in patients with CB (Supplementary Fig. 8c). These exploratory findings suggest that functional and less suppressive CD8 + T-cell subsets may also correlate with clinical response.

## Discussion

The treatment armamentarium for heavily pretreated BRCAwt platinum-resistant HGSOC patients is narrow, and new therapeutic options are needed. ATR/CHK1 pathway inhibitors have been studied as a therapeutic strategy to target cancers with high levels of RS, e.g., platinum-resistant HGSOC[7]. Unlike earlier limited data with ATRi monotherapy[55–57], single-agent CHK1i exhibited promising activity in a subset of platinum-resistant recurrent BRCAwt HGSOC patients (ORR 32%)[29]. Our initial observation led to the multicenter phase 2 trial of CHK1i in recurrent HGSOC which showed mixed findings (ORR 12.1%)[20]. This discordant data prompted us to confirm the clinical activity of CHK1i in the platinum-resistant population and investigate the molecular characteristics of patients who did not derive the benefit from CHK1i. The present study confirmed the previously seen clinical activity of CHK1i in the heavily pretreated platinum-resistant BRCAwt HGSOC population, yielding an ORR of 30.8%, a DCR (PR and SD ≥ 6 months) of 56.4%, and a median PFS of 5 months. However, we could not confirm or refute the role of *CCNE1* amplification as a predictive biomarker because only two patients had tumors demonstrating *CCNE1* amplification[21,58].

Other gene alterations, including amplification/overexpression of oncogenes driving RS and loss of function (LoF) in RS-response-related genes, have been investigated as possible biomarkers for DDR inhibitors, with slightly different gene signatures among the studies[10,22,32]. For instance, ATRi camonsertib resulted in clinical activity in subsets of advanced solid tumors harboring LoF alterations in DDR genes, identified by chemogenomic CRISPR screens[32]. In that study, among 20 heavily pretreated, mostly platinum-resistant, ovarian cancer patients with prior PARP inhibitors (PARPi), four with germinal LoF alterations in *BRCA1* and *RAD51C* genes attained PRs[32]. But further molecular dissection into the replication-fork dynamics, especially involved in DNA replication initiation and replisome progression is needed as new data are emerging[21,59].

In the present study, we identified that genes involving high-fidelity DNA replication initiation and fork progression were associated with CHK1i resistance (*POLA1, POLE, GINS3*). Multiple preclinical studies suggest that CMG helicases components and DNA polymerases may serve as possible therapeutic targets or biomarkers. For example, an RS signature including genes involved in DNA polymerases (*POLA1, POLD4, POLE4*) was found to be predictive for ATRi response in lung cancer preclinical models[59]. Similarly, DNA replication gene knockout

may improve CHK1i activity, by inhibiting *POLA1, POLE*, and *POLE2* genes in a siRNA screen performed on lung and colorectal cancer cells showing low sensitivity to CHK1i[60]. Collectively, those data suggest that the high-fidelity DNA machinery plays a crucial role when backup origin firings are activated during RS to ensure the accurate completion of DNA synthesis[34,35]. It is possible that this error-free DNA replication system may avoid the incorporation of errors into the DNA, and maintain genome integrity, thus tolerating RS[34,35].

Currently, several DNA polymerases are under investigation in the various cancer models[61]. Polymerase eta and beta inhibition were shown to re-sensitize ovarian cancer cells to platinum agents[62,63], and the polymerase theta inhibitor novobiocin appeared to be synthetically lethal with PARPi in in vitro and in vivo models[64,65]. Overall, it is noteworthy that the polymerase beta and theta are selectively toxic for BRCA-deficient tumors[63,65], and our data indicate that *POLA1* inhibition may circumvent CHK1i resistance in the BRCAwt platinum-resistant population, reflecting the complexity of DNA replication-repair interplay in the presence or absence of functional BRCA and the development of drug resistance. Also, Dallavalle et al. reported that a POLA1-HDAC1 dual inhibitor, MIR002, inhibits the primer extension activity of POLA1, resulting in antiproliferative effects on various human cancer cell lines, including ovarian cancer[66]. Despite promising preclinical results, MIR002 has not been tested in humans. More clinical trials investigating the safety and efficacy of these novel DNA polymerase blockades are eagerly awaited.

In addition, we found in the CB group a higher mRNA expression of *POLDIP2*, which is the gene regulating the relative usage of TLS over the template switching pathway upon RS[49] and the significant enrichment of the UV and TLS-associated SBS7d mutational signature. Clinically, Yap et al. reported ATRi camonsertib sensitivity in patients with UV-light-associated mutational signatures[32]. Similarly, we observed a correlation between CB from CHK1i and the RS-induced *GADD45B* gene, which is activated by environmental stresses, including UV[50]. We speculate that mild RS affecting HGSOC may be tolerated by upregulation of the high-fidelity, error-free DNA replication machinery, ensuring DNA replication completion, and leading to CHK1i resistance. In contrast, the bulky DNA damage requires the activation of the error-prone TLS pathway to restart DNA synthesis, resulting in the incorporation of DNA errors[49]. TLS also causes platinum resistance by bypassing platinum-DNA adducts and stress-induced mutagenesis[67–69]. Therefore, we hypothesize that the TLS-induced mutagenesis might cause replication-fork collapse, cell death, and thus CHK1i sensitivity in the absence of the CHK1-dependent checkpoint regulation.

Lastly, we investigated the predictive role of different subsets of CTCs. In recent years, CTCs expressing epithelial markers, such as EpCAM and MUC1, were shown to have a prognostic value, demonstrating a positive correlation with shorter overall survival before surgery and after chemotherapy[70,71], and with chemo-resistance[71]. Findings from our study may also suggest the potential of monitoring the early dynamic changes in the EpCAM+ MUC1+ CTCs during the first 15 days of treatment to predict CHK1i responses, due to the evidence of an improved PFS in patients presenting a decrease in the number of EpCAM+ MUC1+ CTCs. The detection of CTC changes upon CHK1i treatment may represent a non-invasive biomarker approach to identify patients who are likely to benefit from CHK1i therapy, requiring prospective validation in large studies.

Limitations of our study include the proof-of-concept single-arm design of the clinical trial and the small sample size because the trial was stopped early before enrolling the planned numbers due to COVID-19 and the discontinuation of investigational drug supplies. Biomarker findings are based on post hoc analyses without any formal adjustment for multiple comparisons due to the small sample size. Although we have used pre-treatment fresh biopsy samples to reflect best the dynamic nature of RS, further validation is needed in a larger,

prospective setting. The results of the current study, along with other clinical trials, must be interpreted cautiously as they are hypothesis-generating, given the lack of a standardized definition of RS, which may be the promising biomarker of ATRi or CHK1i sensitivity. Lastly, we acknowledge the potential limitations of exploratory findings from analyses in two cell lines to a broader HGSOC population and the lack of mouse experiments, requiring further studies in various preclinical and clinical HGSOC models.

In summary, our study further confirms that CHK1i may be a valuable therapeutic option for the heavily pretreated, BRCAwt platinum-resistant HGSOC patients. Currently, CHK1i prexasertib (a.k.a. ACR-368) is being investigated in the molecularly-selected platinum-resistant HGSOC patients by the proteomics-based biomarker, OncoSignature[72,73] (NCT05548296). Our translational research, along with preclinical models, allows the molecular characterization of the subset unlikely to benefit from CHK1i monotherapy. Transcriptomic profiles and in vitro findings exhibited the involvement of genes related to DNA replication initiation and fork progression in CHK1i resistance. Also, it is noteworthy that DNA replication inhibitors (e.g., POLA1 inhibitors) may represent a potential opportunity to overcome CHK1i resistance in platinum-resistant BRCAwt HGSOC.

## Methods

### Clinical trial design and patient characteristics

49 BRCAwt platinum-resistant recurrent HGSOC patients were enrolled between January 25, 2017 and March 23, 2020. This report describes the final analyses of the two BRCAwt platinum-resistant HGSOC cohorts (cohorts 5 and 6 in the clinical trial protocol, available as Supplementary Note in the Supplementary Information), with biomarker analyses from an open-label, single-arm phase 2 basket trial (NCT02203513, date of study registration on clinicalTrials.gov: July 30, 2014). Originally, six independent cohorts were included in the study, including *BRCA*-mutated HGSOC[22], BRCAwt HGSOC[29], and BRCAwt triple-negative breast cancer patients[74], BRCAwt platinum-resistant HGSOC patients with biopsiable disease (cohort 5) and without biopsiable disease (cohort 6). This is the first and final report of cohorts 5 and 6. The trial was conducted according to federal law and good clinical practice regulations and was approved by the Institutional Review Board of the Center for Cancer Research (CCR), National Cancer Institute (NCI), USA. The trial was designed and conducted according to federal law, good clinical practice regulations, and the Declaration of Helsinki.

Specifically, eligible patients for cohorts 5 and 6 were ≥18 years old with histologically or cytologically confirmed recurrent, platinum-resistant HGSOC, primary peritoneal cancer, and/or fallopian tube cancer who had been previously treated without an upper limit on the number of previous lines of therapy. BRCAwt status was required as a key inclusion criterion: patients must have had a negative germline or somatic BRCA1/2 mutation as determined by a CLIA-certified laboratory. Other inclusion criteria were the following: measurable disease per RECISTv1.1; an Eastern Cooperative Oncology Group (ECOG) performance-status score of 0–2; adequate organ and marrow function. Concerning the biopsiable disease group (cohort 5), all participants must have had at least one lesion deemed safe to biopsy. Key exclusion criteria included prior treatment with prexasertib or other cell-cycle checkpoint kinase inhibitors, concurrent anticancer treatment or any investigational anticancer therapy ≤ 4 weeks before prexasertib, absence of central nervous system metastases ≤ 1 year of enrollment, serious or uncontrolled concurrent illness or infection, history of drug-induced serotonin syndrome.

### Clinical trial procedures

All patients provided written informed consent before enrollment, and eligible patients received intravenous prexasertib monotherapy at 105 mg/m$^2$ every two weeks in 28-day cycles. Patients received treatment until disease progression, unacceptable toxicity, inter-current medical issues, or participant withdrawal of consent. Safety was assessed at each study drug administration at cycle 1, and every 4 weeks for subsequent cycles. AEs were graded according to CTCAEv4.0. Laboratory assessments and electrocardiograms were performed at each cycle, within 24 h before each prexasertib administration. A complete blood count was performed on day 8 of cycle 1 for absolute neutrophil count nadir. CT scans were performed every 2 cycles for RECIST v1.1 evaluation. Dose reduction to 80 mg/m$^2$, and subsequently to 60 mg/m$^2$ every two weeks was required for grade 3 or 4 thrombocytopenia > 7 days or for any grade thrombocytopenia requiring platelet transfusion for bleeding. Any grade neutropenia lasting ≤ 7 days without fever did not require dose reduction or discontinuation of treatment.

### Clinical trial objectives and endpoints

The primary endpoint was ORR as assessed by Investigators according to RECISTv1.1. Secondary endpoints included safety and PFS. The evaluation of pharmacodynamic and predictive biomarkers of CHK1i response or resistance was an exploratory endpoint.

### DNA and RNA sequencing

All 24 patients enrolled in cohort 5 underwent percutaneous needle biopsies by CT or ultrasound guidance at baseline. Biopsy samples were processed immediately in real-time into optimal cutting temperature compound, stored at −80 °C, and then cut and stained immediately before use. The tissue area was measured and prepared to obtain the optimal quality of the tissue, defined as core biopsy samples with solid tissue areas containing ≥ 50% tumor cells and < 25% necrosis[75]. Six of 24 patients' biopsies failed to meet these criteria due to high contents of necrotic cells.

### BROCA-GOv1 and WES

DNA sequencing was performed on pre-treatment core biopsies of 18 patients enrolled in cohort 5 as previously described[76]. Briefly, BROCA-GOv1 of the gene panel was used to detect the alterations in genes listed in Supplementary Data 3. Estimated copies of the gene or exon(s) present in the tumor cells of the sample were calculated by correcting the sample CN with the estimated tumor cellularity and reported in Supplementary Data 4. Three samples had < 15% cellularity thus no downstream analysis was performed by BROCA-Gov1 (Supplementary Data 4).

NovaSeq 6000 S1 sequencing system (Illumina, CA, USA) was used to perform WES at the CCR Sequencing Facility/NCI, to identify genetic variants altering protein sequences. Total DNA prepared from tumor biopsies and matched-normal buffy coats were sequenced. The samples were mapped to the hg38 genome, and variants were called using DRAGEN (v3.9.5). The mapped sequencing depth coverage over the target (after alignment and marking duplicates) ranged from 141x to 395x. The mean insert size for these samples was between 185 and 262 bases. More than 95% of the target region had coverage above 20x. Somatic variants were annotated using VEP (v97)/vcf2maf (v1.6.18) and the subsequent MAF files were used for downstream analysis. Sequenza (v3.0.0) was used for estimating allele-specific CN while adjusting for tumor ploidy and cellularity. All variant calls and CN files were input into R/Bioconductor (v4.2.2) using the maftools (v.2.14.0) package for analysis and visualization. The sigminer (v2.1.9) package was used to extract the SBS, DBS, ID, and CN signatures of the data. Signatures were tallied to the hg38 genome build. Signatures were then extracted using the automatic relevance determination technique using 25 initial signatures, 10 runs, and L1KL method parameters. Given the extracted signatures, the cosine similarities of all signatures were compared to the COSMIC v3.1 reference signatures. Additionally, we fit signature exposures with a linear combination decomposition to the COSMIC v3.1 databases. Across each mutational signature type, we

compared relative signature exposures across PFS groups, testing the differences in signature exposure by PFS status using a non-parametric Kruskal–Wallis test.

## RNAseq

RNAseq was performed using a HiSeq3000 sequencing system (Illumina) at the CCR Sequencing Facility/NCI on pre-treatment biopsy samples from 17 patients enrolled in cohort 5. RNAseq samples were sequenced on NovaSeq S1 using Illumina TruSeq Stranded Total RNA Library Prep and paired-end sequencing. The samples have 86 to 122 million pass filter reads with more than 89% of bases above the quality score of Q30. Reads of the samples were trimmed for adapters and low-quality bases using Cutadapt before alignment with the reference genome (Human - hg38). Public data available from the Cancer Genome Atlas (TCGA) and the Genotype-Tissue Expression portal (GTEx, https://www.gtexportal.org/home/) for ovary and liver (normal and tumor, when available) were used for gene expression studies. Common genes between all datasets were intersected and retained for downstream analysis. Raw counts were then input into a DGElist for analysis. We filtered genes with a cpm <1 in 15 or fewer samples. All data were then batch-normalized using ComBat from the sva R/Bioconductor package. The batch variable used for ComBat was a three-level factor considering the study of origin (TCGA, GTEx, or internal). Principal component analysis was constructed from the batch-normalized data after centering. The first two PCs were plotted across all data with labels for batch and tumor/normal status. Two samples did not match with ovarian cancer tissue profiles, since they were more likely normal or necrotic tissues. 15 biopsies were selected for further in silico analysis. Quartile normalization and log transformation prior to analysis were performed on datasets. GSEA analysis using gene_set-based premutation mode with 5000 permutations was done by GSEA software (v4.3.2).

## CTCs analysis

Peripheral blood samples (8 mL EDTA tubes) were collected at baseline, and at C1D15. After RBC lysis, blood cells were incubated with nuclear dye (#H3570, Hoechst 33342, Life Technologies, DC, USA), viability dye (#L34966, LIVE/DEAD Fixable Aqua, Life Technologies) and antibodies including PE-conjugated anti-human epithelial cell adhesion molecule (EpCAM) Ab (#130-091-253, clone HEA-125, Miltenyi Biotec, CA, USA). The anti-PE magnetic beads (#130-048-801, Miltenyi Biotec) were then used to enrich EpCAM-positive cells. Cell quantification was calculated by multiparameter flow cytometry[77–79]. Viable, nucleated, EpCAM-positive, CD45 (#304014, clone HI30, BioLegend, CA, USA) negative cells were finally considered CTCs and further characterized for CD117 (#313212, clone 104D2, BioLegend), CXCR4 (#306516, clone 12G5, BioLegend), PDL1 (#329708, clone 29E.2A3, BioLegend) and MUC-1 (#559774, clone HMPV, BD Biosciences, CA, USA) expression. Antibody dilution details are provided in Supplementary Table 4.

## Immune cell subset analysis

Peripheral blood specimens (two 8 mL BD Vacutainer CPT tubes) were collected at baseline, and at C1D15. Peripheral blood mononuclear cells (PBMCs) were obtained using centrifugation and viably frozen until analysis. PBMCs were incubated with Fc receptor blocking reagent (Miltenyi Biotec) and stained with monoclonal antibodies (20 min at 4 °C). Dead cells were excluded from the analysis using the viability dye, LIVE/DEAD Fixable Aqua. All analyses were performed using multiparametric flow cytometry (MACSQuant; Miltenyi Biotec). Data were analyzed using FlowJo software v.10.6.1 (FlowJo, LLC, OR, USA). Cells were gated on specific immune cell subsets (Supplementary Fig. 9-12), and further for functional markers (Supplementary Table 5). The monoclonal antibodies used (all from BioLegend) are listed in Supplementary Table 4.

## Cell lines

OVCAR3 and OVCAR5 (BRCAwt platinum-resistant HGSOC cell lines) were obtained from NCI-60 collection at the NCI, Frederick, MD, USA. The CHK1i-resistant OVCAR5-PrexR cell line was developed from parental OVCAR5 as described earlier[45]. OVCAR3-PrexR cell line was a gift from Dr. Michail Shipitsin, Acrivon Therapeutics Inc., based on the Material Transfer Agreement between Acrivon Therapeutics and NCI. All cell lines were cultured in RPMI-1640 with medium L-glutamine (Life Technologies) and supplemented with 10% FBS, 1% penicillin/streptomycin, 1 mM sodium pyruvate, and 5 μg/mL of insulin from bovine pancreas (Sigma-Aldrich). Authentication was evaluated by short tandem repeat analysis conducted by Labcorp (NC, USA) and tested negative for mycoplasma using MycoAlert (#NC9719283, Lonza).

## Cell growth assays

Viable cells were counted and plated at 4,000 cells per well in triplicate in flat-bottomed 96-well plates and incubated overnight, before treating with specific inhibitors. After 48 h of treatment with the inhibitors, growth inhibition was assessed with XTT reagent as per the manufacturer's instructions (Thermofisher Scientific, Waltham, MA, USA). The absorbances were measured at 490 nm on a BioTek SynergyHT™ plate reader using Gen5™ software (BioTek Instruments) and analyzed on Microsoft Excel.

## siRNA transfection

ON-TARGETplus SMARTpool-Human of four specific POLA1 (#L-020856-00-0005) and POLE (#L-020132-00-0005) siRNAs (Horizon Discovery Lafayette, CO, USA) were used to transiently transfect cells at 70% of confluence with Dharmafect 1 reagent (Horizon Discovery) as per manufacturer's protocol. Non-targeting control siRNA (#D-001810-10-20) were used as negative controls. After 24 h, cells were trypsinized and plated at the required density into 96-well plates (for XTT assays) or 6-well plates (for immunoblot assays). The next day cells were treated with appropriate inhibitors and incubated for 48 h for XTT assays or overnight prior to cell harvest and protein lysate preparation for immunoblots. The sequences of siRNAs used in this study are listed in Supplementary Data 8.

## Immunoblotting

Immunoblotting was performed as described[80]. Briefly, protein extracts were prepared in RIPA buffer (Thermofisher Scientific) supplemented with protease (complete™) and phosphatase inhibitors (PhosSTOP)™ (Roche Diagnostics, Indianapolis, IN, USA). Proteins were separated on 4-12% Bis-Tris gels (Novex™, Thermofisher Scientific) and transferred onto 0.45 μ PVDF membranes (Millipore) using the XCell II® Blot module (Thermofisher Scientific). Blots were blocked with 5% BSA in TBS-T (0.05% Tween 20 in TBS), treated with specific antibodies, followed by HRP-conjugated secondary antibodies. Blots were visualized with Supersignal® West Dura extended duration substrate (Thermofisher Scientific) and documented on an Odyssey™ Fc gel documentation system (LI-COR biosystems, NE, USA). Densitometric analysis was performed using ImageStudio™ software (LI-COR biosystems). Antibodies against total CHK1 (#2360), CHK1-S345 (#2348), CHK1-S296 (#2349), pKAP1-S824 (#4127), total KAP1 (#5868), β-Actin (#3700), anti-mouse IgG HRP (#7076) and anti-rabbit IgG HRP (#7074) were from Cell Signaling Technology (Danvers, MA, USA), while antibodies against POLA1 (#Ab31777) was from Abcam (Waltham, Massachusetts, USA) and those against POLE (#PA5-78113) were obtained from Thermofisher Scientific. Antibody dilution details are provided in Supplementary Table 4.

## Statistical analyses

Both cohorts 5 and 6 utilized a single-stage phase 2 design with an early stopping rule. For cohort 5, the sample size of 36 patients was selected

to rule out an ORR of 20% in favor of an ORR of 45%, using a two-tailed $\alpha = 0.05$ for an 89% power. The sample size of 35 patients for cohort 6 was chosen to rule out an ORR of 15% in favor of an ORR of 40%, using two-tailed $\alpha = 0.05$ for an 88% power. The regimen would be considered sufficient for the next stage of clinical development if ≥13/36 patients had a CR or PR in cohort 5, with the exact two-sided 95% CI ranging from 20.8–53.8%, surpassing the minimum 20% ORR and containing the target 45% ORR. Conversely, the data would be sufficient if ≥11/35 patients achieved CR or PR in cohort 6, with the exact two-sided 95% CI ranging 16.9–49.3%, surpassing the minimum 15% ORR and containing the target 40% ORR. The accrual would be stopped if no one in the first 10 enrolled patients had CR or PR in either group, as the upper bound on a one-sided 90% CI of 1/10 patients would have been 33.7%. There was a high probability that the true ORR could be less than 33%, which was obtained in a previous cohort of recurrent BRCAwt HGSOC[29]. Safety analyses included all patients.

Descriptive statistics (median, frequencies, ranges) were used to summarize baseline characteristics, AEs, and response measurements (ORR, DCR). 95% of CIs for ORR and DCR were analyzed using the Clopper-Pearson method (R Studio, Version2022.12.0 + 353). Median PFS was calculated using the Kaplan–Meier method (R Studio, Version2022.12.0 + 353). Fisher's exact test (two-sided) and Welsh Two Sample TOST (two one-sided tests) were used to compare the proportions of clinical outcomes between cohorts 5 and 6. Of note, the study was prematurely terminated because the company withdrew the sponsorship for investigational drug supplies in February 2020. Because cohorts 5 and 6 were closed early for enrollment, the participants from cohorts 5 and 6 were reported together as a combined dataset to have a reasonable number of data analyses given the results are sufficiently similar between the two cohorts.

For statistical analyses of RNAseq, multiple hypothesis testing followed by adjusting with Benjamini–Hochberg false-discovery rate (FDR) cut-off <10% (FDR $q < 0.1$ indicates significance) was used. For GSEA, FDR $q < 0.25$, |normalized enrichment score (NES)| >1.65, and nominal $P < 0.02$ were used as selection criteria[81]. Comparison of mutational signatures was performed using a Kruskal–Wallis test and grouped P-values according to mutational signature-related etiology were obtained using a Fisher exact method (WES analysis). Comparison of immune cell subsets and CTCs was calculated by a non-parametric Wilcoxon rank sum test for unpaired samples, while the Wilcoxon matched-pairs test was used to analyze paired samples. All differences were considered statistically significant if $P < 0.05$. PFS according to CTCs changes was estimated by the Kaplan–Meier method; comparisons between arms, hazard ratio, and CI were analyzed using the log-rank test.

For preclinical studies, at least duplicate independent biological replicates were performed in all experiments. Investigators were blinded during data collection and analysis. Data were analyzed using a standard Student's $t$-test (two-sided) to determine significance and are shown as mean ± standard deviation (SD). P-values < 0.05 were considered significant. All statistical analyses were done using GraphPad Prism v9 or Microsoft Excel.

### Reporting summary
Further information on research design is available in the Nature Portfolio Reporting Summary linked to this article.

## Data availability
The DNA sequencing data of BROCA-GOv1 generated in this study have been deposited in the BioProject database under accession code PRJNA1087413. The WES data generated in this study have been deposited in the dbGaP database under accession code phs003588.v1.p1. Access to the WES raw data requires dbGAP authorization, so as to provide oversight and investigator accountability for potentially sensitive datasets involving the study subject's health information. The RNAseq data generated in this study have been deposited in the GEO database under accession code GSE249587. The processed RNAseq data are available at Supplementary Data 5. The raw clinical data are protected and are not available due to data privacy laws. Specific requests for access to de-identified clinical data should be sent to the corresponding author. The study protocol is available as a Supplementary Note in the Supplementary Information. The publicly available data of mRNA alteration of *POLE* and *POLA1* used in this study are available in the cBioPortal database [https://www.cbioportal.org][82–84]. The publicly available data of mRNA alteration of *POLE* and PFS in HGSOC used in this study are available in the Kaplan–Meier Plotter [ovarian cancer] database [http://www.kmplot.com][85]. The remaining data are available within the Article, Supplementary Information, or Source Data file. Source data are provided in this paper.

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

## Acknowledgements

The authors thank Dr. Chen X, Mrs. Bao T, and Mrs. Shetty J at the Sequencing Facility, NCI at Frederick; Dr. Cam M, Lack J, and Abdalla A, at the Center for Cancer Research (CCR) Collaborative Bioinformatics Resource, NCI for their expertise in generating the RNAseq datasets. We also thank C. Annunziata, E. Kohn, E. Curreri, M. Gomez, J. Presentacion Blanco, E. Villanueva, M. Pavelova, B. Solarz, and A. Zimmer for contributions to the clinic; S. Steinberg for assistance in statistical analysis of the clinical trial and all the patients enrolled in the trial and their caregivers. This research work was funded by the Intramural Research Program of the Center for CCR, NCI, NIH (grant ZIA BC011525 awarded to J-M.L.), the Department of Defense Investigator-initiated Research Award (OC160355 [E.M.S]). Prexasertib was supplied to the NCI CCR by Eli Lilly under a cooperative research and development agreement.

## Author contributions

Study concept and design: E.G., J.R.N., G.Z., and J.M.L. patients' enrollment and treatment, acquisition of data: A.M., S.L., and J.M.L.; analysis and interpretation of data: E.G., J.R.N., G.Z., T.T.H., D.N., K.I., D.D., T.M., E.M.S., M.R.R., M.J.L, B.R., E.L, S.R., N.S., J.B.T., and J.M.L.; statistical analysis: E.G., J.R.N., T.T.H., D.D., T.M., and D.N.; drafting of the manuscript: E.G., G.Z., J.R.N., T.T.H., and J.M.L.; manuscript review: all authors.

## Funding

## Competing interests

J.-M.L. has research grant funding from AstraZeneca and Acrivon Therapeutics (paid to institution) and is on the Scientific Advisory Board of Acrivon Therapeutics and Genentech (unpaid). E.M.S. is on the Data and Safety Monitoring Board of Novartis and the Scientific Advisory Board of Ideaya Bioscience. The other authors have no competing interests to declare.

## Additional information

[1]Women's Malignancies Branch, Center for Cancer Research (CCR), National Cancer Institute (NCI), National Institutes of Health (NIH), Bethesda, MD 20892, USA. [2]Institute of Obstetrics and Gynecology, Università Cattolica del Sacro Cuore, Largo Agostino Gemelli 8, 00168 Rome, Italy. [3]Center for Cancer Research Collaborative Bioinformatics Resource, CCR, NCI, NIH, Bethesda, MD 20892, USA. [4]Department of Ob/Gyn, University of Washington, Seattle, WA 98195, USA. [5]Statistical Consulting and Scientific Programming Group, Computer and Statistical Services, Data Management Services, Inc. (a BRMI company), NCI, Frederick, MD 21702, USA. [6]Developmental Therapeutics Branch, CCR, NCI, NIH, Bethesda, MD 20892, USA. [7]Clinical Image Processing Service, Department of Radiology and Imaging Sciences, CCR, NCI, NIH, Bethesda, MD 20892, USA. [8]Interventional Radiology, CCR, NCI, NIH, Bethesda, MD 20892, USA. [9]These authors contributed equally: Elena Giudice, Tzu-Ting Huang, Jayakumar R. Nair. ✉e-mail: leej6@mail.nih.gov

