## [Peer Review File · Nature Communications]

The CHK1 inhibitor prexasertib in BRCA wild-type platinum-resistant recurrent high-grade serous ovarian carcinoma: a phase 2 trialREVIEWER COMMENTS

Reviewer #1 (Remarks to the Author): with expertise in biostatistics, clinical trial study design

Overview: The study reported two BRCA wild type platinum-resistant HGSOC cohorts (cohort A: with biopsiable disease and cohort B: without biopsiable disease) from an open-label, single-arm phase 2 trial. Biomarker analysis was also included for various types of omics data.

Statistical comments: There are significant issues in sample size and data analysis.

1. Cohort A and B planned to have 36 patients and 35 patients, respectively. However, the report had n=24 for Cohort A and n=25 for Cohort B. There is no explanation for the difference.
2. Results of safety and efficacy is lack of details. For example, safety data was presented in Table 2 with little explanation in the main text. Data for duration of response was also not provided.
3. The hypothesis is different in both cohorts with the null hypothesis of 20% ORR in Cohort A and 15% ORR in Cohort B. Stratification by cohort is more appropriate for analysis of response data to account for cohort heterogeneity. It also helps assess if ORR is different between the two cohorts prior to pooling both cohorts. Other clinical outcomes have the same issues (e.g., DCR and PFS).
4. One critical problem in the biomarker analysis is lack of consideration of false discovery issue given exploration of numerous omics features. Many results with $p=0.01-0.05$ are likely to be non-significant after adjustment of multiple testing (e.g., POLA1 ($P=0.04$) and POLE ($P=0.03$) genes in transcriptomic profile).
5. It would be more informative to compare the two cohorts in biomarker analysis, as well as subgroup analysis stratified by cohort, to evaluate the biopsiable disease effect, which is one key factor for the design of the two cohorts.

Reviewer #2 (Remarks to the Author): with expertise in ovarian cancer, therapy

The authors present analyses of samples from a clinical trial with a CHK1 inhibitors (CHK1i)

in patients with ovarian cancer. Some issues to consider include:

- 1) Line 3: Presumably, the authors are referring to approximately 25% of patients who have either germline or somatic BRCA1 or 2 mutations?
- 2) Line 65. Please check wording – HGSOCS don't necessarily have "universal loss of p53" but rather near universal presence of mutations in p53; many of the mutations don't result in "loss of p53".
- 3) Line 101. Do the patients presented in the current paper include ones presented in the prior publication by the authors (Lee et al., Lancet Onc 2018) or is this a completely different trial? If this is a completely different trial/set, then would state that clearly. Also, what was the exact goal of this trial compared to the previous one? There have been other phase 2 trials already reported with this drug as well (e.g., Konstantinopoulos et al., 2022).
- 4) Line 138. In this section, why use clinical benefit since it's not an endpoint that could be used for drug approval? Is a PFS of 5.5 months (< 6 months) really all that different than PFS of 6.5 months (PFS >6 months)? Why not use ORR?
- 5) Line 151. What was the depth of sequencing with WES?
- 6) Line 184. How were the doses of aphidicolin selected? What doses are effective for inhibiting polymerases? Doses used here seem very high.
- 7) Line 186. Here and in other places, the term "synergistic" is used but the data shown are mostly limited inhibition studies. For assessment of synergy, formal assessment (e.g., Combination Index) should be assessed and calculated.
- 8) Line 202. As above, formal assessment of synergy should be done.
- 9) Line 193. In this section, apparently siRNA pools were used. How was specificity of siRNA assessed? What was the extent of knockdown of the genes of interest?
- 10) Page 8. It would be important to show in vivo data; otherwise the data are not particularly convincing. Ideally this would be done in several models and also assess tissue-level effects.
- 11) Page 7. There are platinum-sensitive and platinum-resistant versions of the OVCAR3 and OVCAR5 cells reported. What is the platinum IC50 level of the cells that the authors are using?
- 12) Page 11. Portions of the Discussion simply repeat the results. Would suggest re-doing the discussion to focus more on interpreting the results and presenting context rather than

repeating results.

Reviewer #3 (Remarks to the Author): with expertise in ovarian cancer, therapy

The authors herein report the results of an important effort of translational research regarding the activity of prexasertib, a CHK1 inhibitor, in platinum-resistant high grade epithelial ovarian cancer (PROC).

It is noteworthy that this team has already published the results of the first proof of concept phase 2 study of prexasertib in *Lancet Oncol*, 2018. This preliminary assessment was conducted on 28 patients, of whom 24 were assessable. ORR was 33% in the treated population. In a much larger study published in *Gynecologic Oncology* (2022), Konstantinopoulos et al included 169 patients with PROC and found an ORR of 12.1 %. In 2018, Eli Lilly decided to stop the development of prexasertib and the drug was sold to Acrivon in 2021.

In the present manuscript, the authors tried to find new biological predictive factors of prexasertib efficacy through multiple analyses of genomic and transcriptomic profiling, mutational signature, in vitro studies of cancer cell lines, and CTC / immune cells quantification and characterization.

The manuscript is well-written and the methodology of biological analyses is adequately described.

MAJOR COMMENTS

1) Despite multiple tests, it seems that the only pertinent finding in this study was the upregulation of genes involved in the DNA high-fidelity replication machinery, among which POLA1 could represent a potential target for future drugs, but which ones ?

To our knowledge, MIR002 could be a candidate [Dallavalle S, et al. Antitumor activity of novel POLA1-HDAC11 dual inhibitors. *Eur J Med Chem*. 2022, 228:113971.] but has never been tested in humans. This should be discussed by the authors.

2) This line of research seems to have been abandoned by Acrivon, which has adopted the OncoSignature companion test in its current basket study in gynecological tumors. The OncoSignature companion tests uses mass spectrometry, biased tumor model analyses, and quantitative multispectral in situ imaging of patient-derived xenografts to measure protein biomarkers and other biological mechanisms that are driving patients' tumors and can be interrogated by targeted drugs. The authors mentioned this approach in their conclusion but it should be further discussed.

3) One could be surprised by the number of patients who were finally not evaluable for response (10/49), which is unexpectedly high (13/169 in the pivotal study by Konstantinopoulos). Three patients withdraw their consent: was it due to toxicity ? One could imagine that patients with the poorest prognosis (sepsis, renal failure, fistula, etc.) could not complete the treatment. Thus, the ORR reported in evaluable but highly selected patients seems overestimated, which casts doubt on the translational analysis itself. Are some descriptive results available in the 10 "non assessable patients" and how do they compare to the overall results presented in the paper ?

4) According to their statistical hypothesis, the authors should have included 71 patients in their trial. Only 39 were finally evaluable. Therefore, the study must be considered as inconclusive. It is difficult to interpret the results of ancillary studies in this context. Thresholds for significance are borderline and multiple tests do not allow to avoid results which would have been obtained by chance.

OTHER COMMENTS

Why were BRCA mutations excluded from the trial, since the BRCAm status has no particular prognostic influence in heavily pretreated PROC patients ? The sentence in the introduction l.60-61 should be modified : the unmet need is for both categories of patients.

The distinction between primary and secondary platinum resistance is artificial and is not based of current evidence nor clinical guidelines. Fig 2.e shows nos difference in terms of response and PFS.

The authors should discuss the discrepancies between their study and the results presented in Gynecol Oncol, because it seems unlikely that they could be based on biological heterogeneity only.

Reviewer #4 (Remarks to the Author): with expertise in ovarian cancer, therapy

Synopsis of findings:

Clinical outcomes of the study are reported. They report an ORR of 30.7% in the RECIST-evaluable population with a median PFS of 5.8 months. This is a clinically meaningful result in this challenging to treat patient population.

No correlation between CCNE1 amplification and clinical response to prexasertib was identified. It is noted that only 2 patients had tumors demonstrating CCNE1 amplification, so it is perhaps more appropriate to say that this study could not meaningfully confirm or refute the role for CCNE1 amplification as a predictive biomarker.

The authors present data from NGS and whole exome sequencing. No significant trends or predictors are identified.

The authors conducted transcriptomic analysis and identified a significant mRNA overexpression of POLA1 and POLE in the patients without clinical benefit, as well as overexpression of MCM7 and GINS3. The authors hypothesize that upregulation of high-fidelity polymerases (POLA1, POLE) may be critical for DNA synthesis in BRCAwt platinum-resistant HGSOC.

They subsequently tested this hypothesis in cell lines (OVCAR3 and OVCAR5, platinum-resistant BRCAwt HGSOC cell lines) using a pan-inhibitor of B-family polymerases. The combination of a pan-inhibitor of B-family polymerases with prexasertib demonstrated synergistic cell growth inhibition.

siRNA targeting POLA1 and POLE were used to identify the respective role of each. POLA1

silencing resulted in inhibition of growth in both OVCAR3 and OVCAR5 cell lines.

They subsequently evaluated mutational signatures and found significantly higher SBS7d and ID13 mutational signatures in the CB group compared to the NCB group. They propose that these cells benefiting from CHK1i may be more dependent on the error-prone translesion synthesis pathway. Notably, transcriptomic analysis did not reveal upregulation of translesion synthesis polymerase genes.

They examined changes in circulating tumor cells and systemic immune cells. No definitive conclusions were drawn from CTC analysis. An increase in myeloid-derived suppressor cells and classical monocytes was observed following exposure to prexasertib in the non-clinical benefit group.

Significance to the field:

Therapies are greatly needed for patients with platinum-resistant recurrent, HGSOE. The results of this phase 2 trial represent a meaningful clinical result in this population.

As the authors point out on multiple occasions, there is considerable molecular and genetic heterogeneity in the HGSOE population. This problem is only compounded by the complex and dynamic nature of replication stress response. This work is an ambitious undertaking to identify potential predictive biomarkers to CHK1 inhibition.

Several novel findings are reported, including the potential role of POLA1 and POLE in the response to CHK1 inhibition.

Specific comments to authors:

This study is limited to patients with BRCA wild type disease. The rationale for this limitation is presented in the discussion but could be more clearly presented in the introduction.

Line 213 – note typo in the spelling of prexasertib

By performing exploratory analyses in cell lines, albeit 2 different lines, many of the findings presented here may not be reproducible in a broader HGSOC population. Further research to address this consideration would be warranted, and the need for further investigation into these findings is generally proposed. I would recommend making more explicit the limitations in the application of these findings to the broader landscape of HGSOC.

Methodology:

The methodology appears to be rigorous, both in the execution of the clinical trial reported and in the subsequent hypothesis-generating preclinical studies.

Reviewer #1 (Remarks to the Author): with expertise in biostatistics, clinical trial study design

Overview: The study reported two BRCA wild type platinum-resistant HGSOC cohorts (cohort A: with biopsiable disease and cohort B: without biopsiable disease) from an open-label, single-arm phase 2 trial. Biomarker analysis was also included for various types of omics data.

Statistical comments: There are significant issues in sample size and data analysis.

1. Cohort A and B planned to have 36 patients and 35 patients, respectively. However, the report had n=24 for Cohort A and n=25 for Cohort B.

Response: We would like to clarify that we could not enroll the planned number of patients because the trial was prematurely terminated due to the unforeseen circumstances such as COVID19 and Eli Lilly's withdrawal of sponsorship of supplies of the investigational drug prexasertib in early 2020. Therefore, the participants from the cohorts A and B are being reported together as a combined dataset to have a reasonable number for data analysis given that the clinical results were sufficiently similar between the two cohorts (**New Supplementary Table 1** and see the responses to the comments # **2** and **3**). This information has been added to the Results section (**page 5**) and to the study limitation in the Discussion section (**page 13**) for clarity.

Also, we have submitted the new clinical protocol amendment to explain the reasons why cohorts A and B were closed early before enrolling the planned numbers (see NIH IRB approved-clinical protocol version date 11/13/2023 in the **Supplementary Information**, specifically the pages 2 and 67 of the protocol, cohorts 5 and 6 in the protocol = cohorts A and B in the manuscript).

2. Results of safety and efficacy is lack of details. For example, safety data was presented in Table 2 with little explanation in the main text. Data for duration of response was also not provided.

Response: We have expanded the safety data explanation in the main text (**page 5**) as suggested. Briefly, the most common (in >10% patients) grade 3 or 4 treatment-related adverse events (TRAEs) were hematological toxicities, such as neutropenia (42/49, 85.7%), leukocytopenia (38/49, 77.6%), lymphocytopenia (23/49, 46.9%), thrombocytopenia (20/49, 40.8%), anemia (15/49, 30.6%) and febrile neutropenia (6/49, 12.2%), consistent with previous reports.

With respect to the median duration of response (DoR), of 12 patients who achieved the partial response (PR) per RECIST criteria, median DoR was 5 months (95% CI 3-11). We have provided the updated clinical data in the **new Supplemental Table 1** for the combined dataset as well as for each cohort and updated the manuscript accordingly (**page 5**).

3. The hypothesis is different in both cohorts with the null hypothesis of 20% ORR in Cohort A and 15% ORR in Cohort B. Stratification by cohort is more appropriate for analysis of response data to account for cohort heterogeneity. It also helps assess if ORR is different between the two cohorts prior to pooling both cohorts. Other clinical outcomes have the same issues (e.g., DCR and PFS).

Response: Thank you for your suggestion. The reason why we chose a higher cutoff of 20% ORR for null hypothesis for cohort A was because cohort A shared the similar clinical characteristics such as biopsiable disease and recurrent platinum-resistant BRCAwt HGSOc as the previously reported cohort 2 (biopsiable disease and recurrent BRCAwt HGSOc) patients¹. While cohort B represents the same recurrent platinum-resistant BRCAwt HGSOc patients population like cohort A, it was unknown whether non-biopsiable disease such as peritoneal carcinomatosis might negatively play a role to clinical outcome. We therefore elected a more conservative lower cutoff of 15% ORR for null hypothesis for cohort 6.

As per the Reviewer's suggestion, we have conducted a new analysis in collaboration with NCI statisticians. Briefly, a Fisher's exact test comparing treatment response proportions between the two cohorts resulted in a p-value of 1 and an odds ratio of 1.24, indicating strong statistical equivalence. Additionally, the Welsh Two Sample TOST rejects the null hypothesis of differences in best response % among RECIST-evaluable patients (n=39) with a p-value of 8.687318e-14.

Despite the anticipated slight variation in ORR (cohort A: 6/18, 33.3%; cohort B: 6/21, 28.6%) due to the small sample size, both cohorts consistently rejected the null hypothesis.

Progression-free survival (PFS) analysis via Welsh Two Sample TOST ($p=0.69$) and regular t-test ($p=0.20$) also suggests no significant difference between cohorts A and B. Similarly, clinical benefit analysis using Fisher exact test yields a non-significant p-value of 0.20, indicating no significant difference between the cohorts A and B.

We also added the disease control rate (DCR, defined as PR + SD \geq six months in our study) as suggested by the Reviewer. DCR was not significantly different between the two cohorts (Fisher's exact test, $p = 0.20$), with a slight variation (cohort A: 8/18, 44.4%, cohort B: 14/21, 66.7%) likely due to the small sample size. Also, it is noteworthy that there is no clear consensus on the DCR definition in platinum-resistant ovarian cancer like RECIST or PFS. For instance, DCR was defined as PR + SD \geq 6 weeks², 12 weeks³ or four months⁴ depending on the biological and clinical characteristics. We indeed set a high bar of DCR by using 6 months as a cutoff to present more clinically meaningful data for this difficult-to-treat patient population. Regardless, we agree that all subset analyses should be interpreted with a caution and as hypothesis-generating.

These data are summarized in the **new Supplementary Table 1**.

4. One critical problem in the biomarker analysis is lack of consideration of false discovery issue given exploration of numerous omics features. Many results with $p=0.01-0.05$ are likely to be non-significant after adjustment of multiple testing (e.g., POLA1 ($P=0.04$) and POLE ($P=0.03$) genes in transcriptomic profile).

Response: Thank you for your comment. We have performed multiple hypothesis testing followed by adjusting with Benjamini-Hochberg false discovery rate (FDR) cut-off $< 10\%$ (FDR $q < 0.1$ indicates significance). Analysis of bulk RNAseq exhibited no individual gene differentially expressed between the clinical benefit and non-clinical benefit groups, possibly due to the small sample size⁵ (FDR $q = 0.87-1$, **new Supplementary Table 7**). We have added this information to the legend of **new Fig. 3a**.

We also employed diverse methodologies for exploratory analyses in collaboration with NCI statisticians given the limitation of the small sample size. This approach included Gene Set Enrichment Analysis (GSEA)⁶ and the identification of differentially expressed genes with clinical significance^{7,8}. Below details our stepwise approaches for the exploratory analyses.

1) First, we performed GSEA using the Kyoto Encyclopedia of Genes and Genomes (KEGG) database⁶ to identify enriched biological pathways in patients with non-clinical benefit (NCB, PFS < 6 months) versus clinical benefit (CB, PFS ≥ 6 months). FDR $q < 0.25$, $|\text{normalized enrichment score (NES)}| > 1.65$, and nominal $P < 0.02$ were used as selection criteria⁸. Notably, the DNA replication pathway (rank 2, FDR $q = 0.03$, $|\text{NES}| = 1.81$, and nominal $P = 0.003$) was significantly enriched in NCB (**new Fig. 3a** and **Supplementary Table 6**).

2) For further analysis, 969 differentially expressed genes (raw $p < 0.05$ without correction, **new Supplementary Table 7**) were extracted from the gene sets (core enrichment: YES). *POLE* (raw $p = 0.01$) and *POLAI* (raw $p = 0.04$) were significant among 15 genes in the core enrichment of DNA replication pathway (**new Fig. 3b** and **new Supplementary Table 7**).

3) We used the public databases to examine the clinical relevance of *POLE* and *POLAI* in ovarian cancer. In HGSOc, 6.7% (20/300) exhibited the elevated levels of *POLE* mRNA, and 6.7% (20/300) showed high expression of *POLAI* mRNA (**new Supplementary Fig. 2a**). Elevated *POLE* mRNA levels correlated with worse PFS (median 15.0 months vs. 18.2 months; log-rank $p=0.012$, HR=1.36 [1.07-1.74]) in platinum-treated HGSOc patients (**new Supplementary Fig. 2b**). These findings suggest a possible association of *POLE* with clinical outcomes in HGSOc.

4) Lastly, we further investigated the biological significance of *POLAI* and *POLE* in CHK1i sensitivity. Data exhibited that *POLAI* and *POLE* knockdown resensitized the CHK1i-resistant ovarian cancer cell lines to CHK1i (Fig. 4e-f), suggesting that silencing *POLAI* or *POLE* may predispose HGSOc cells to CHK1i treatment.

In the current study, we acknowledge that our biomarker analysis findings must be construed as hypothesis-generating and require testing and validation in the large prospective biomarker-focused clinical trials with an appropriately powered sample size.

The Results and Discussion sections have been updated accordingly (**pages 6-7** and **page 12**). FDR q values for genes and pathways have been incorporated into **new Supplementary Tables 6-7**.

5. It would be more informative to compare the two cohorts in biomarker analysis, as well as subgroup analysis stratified by cohort, to evaluate the biopsiable disease effect, which is one key factor for the design of the two cohorts.

Response: We appreciate your suggestion and have conducted new biomarker analyses (**new Supplementary Fig. 7-8**). Of note, by doing a subgroup analysis stratified by the cohort, the difference between pre and on-treatment for immune subsets is no longer significant. It is possible that reduced sample size and the imbalance between the two cohorts as to the number of clinical benefit group (PFS \geq 6 months, cohort A [n=6] and cohort B [n=12]) vs. non-benefit group (PFS <6 months, cohort A [n=12] and cohort B [n=9]) may have influenced this subgroup analysis.

Reviewer #2 (Remarks to the Author): with expertise in ovarian cancer, therapy

The authors present analyses of samples from a clinical trial with a CHK1 inhibitors (CHK1i) in patients with ovarian cancer. Some issues to consider include:

1. Line 3: Presumably, the authors are referring to approximately 25% of patients who have either germline or somatic BRCA1 or 2 mutations?

Response: Correct. In the Cancer Genome Atlas (TCGA) database, *BRCA1* and *BRCA2* displayed germline mutations in 9% and 8% of ovarian cancers, respectively, with an additional 3% of cases showing somatic mutations in both genes⁹. Similarly, Pennington *et al.* reported that germline mutations in *BRCA1* and *BRCA2* were present in 13.4% and 4.6% of ovarian cancers, respectively,

with somatic mutations in *BRCA1* (5.2%) and *BRCA2* (1.6%)¹⁰. We have updated the references (# 3 and #4 in the manuscript).

2. Line 65. Please check wording – HGSOCS don't necessarily have “universal loss of p53” but rather near universal presence of mutations in p53; many of the mutations don't result in “loss of p53”.

Response: We have revised this sentence (**page 3**) as follows (changes are shown in bold and *italics*): “HGSOCS cells have changes in many genes involved in DNA replication, *e.g.*, universal ~~loss of~~ *TP53 mutations* with consequently defective G1/S cell cycle regulation”.

3. Line 101. Do the patients presented in the current paper include ones presented in the prior publication by the authors (Lee et al., Lancet Onc 2018) or is this a completely different trial? If this is a completely different trial/set, then would state that clearly. Also, what was the exact goal of this trial compared to the previous one? There have been other phase 2 trials already reported with this drug as well (e.g., Konstantinopoulos et al., 2022).

Response: Thank you for your question. This phase 2 study is the NCI single-center investigator-initiated basket trial of CHK1i prexasertib (NCT02203513) (see clinical trial protocol in the Supplementary Information file, page 2, Study Schema).

For Lancet Oncology paper published in 2018¹, we have reported the clinical outcome and correlative studies of the original BRCAwt cohort (cohort 2 in the protocol). Based on the promising activity of CHK1i seen in the original BRCAwt cohort 2, the new BRCAwt cohorts were added to the same NCI single-center basket trial (NCT02203513) to conduct more comprehensive biomarker analyses and to confirm prexasertib activity in platinum-resistant patients. The present report is based on the new BRCAwt cohorts of the NCI single-center phase 2 basket trial (NCT02203513) which includes 49 BRCAwt platinum-resistant recurrent HGSOCS patients enrolled between January 2017 and January 2020 (cohorts 5 and 6 in the protocol [defined as cohorts A and B in this study]).

As a result of promising data from the original BRCAwt cohort of our NCI single-center basket study, Eli Lilly separately opened the multi-center phase 2 basket trial of prexasertib in ovarian cancer as a company-sponsored study (NCT03414047) which was published in *Gynecologic Oncology* in 2022 (Konstantinopoulos *et al.*⁴). We have updated the manuscript to clarify the difference (**page 4**).

4. Line 138. In this section, why use clinical benefit since it's not an endpoint that could be used for drug approval? Is a PFS of 5.5 months (< 6 months) really all that different than PFS of 6.5 months (PFS >6 months)? Why not use ORR?

Response: We agree with the Reviewer that there is no clear consensus on the definition of clinical benefit unlike RECIST criteria. The use of PFS cutoff either 5.5 months or 6.5 months to assess clinical benefit is an interesting idea and should be tested in the future clinical studies. For this report, we chose a cutoff of 6 months of PFS to represent clinically meaningful benefit in heavily pretreated platinum-resistant BRCAwt HGSOc patients because PFS is usually 3-4 months to single agent chemotherapy in this population^{11,12}. We updated the Results sections accordingly (**page 6**).

In addition, this NCI study was intended to investigate the early clinical activity of CHK1 inhibitor prexasertib with comprehensive biomarker analysis. For drug approvals, a separate registration-intent phase 2 basket study of prexasertib (a.k.a. ACR-368) is currently ongoing for platinum-resistant ovarian cancer and uses ORR as a primary endpoint (NCT05548296).

5. Line 151. What was the depth of sequencing with WES?

Response: The sequencing depth coverage over the target region, after alignment and duplicate marking, ranged from 141x to 395x. The mean insert size for these samples was between 185 and 262 bases, with more than 95% of the target region having coverage above 20x. We have updated the Methods (**page 16**).

6. Line 184. How were the doses of aphidicolin selected? What doses are effective for inhibiting polymerases? Doses used here seem very high.

Response: 15 μM concentration for aphidicolin treatment was chosen to aim to strike a balance between minimal toxicity and achieving complete polymerase inhibition in our assays. This decision was guided by previous studies on ovarian cancer cells, demonstrating minimal toxicity up to 25 μM ^{13,14}. The optimization data, assessed through XTT assays for OV90, OVCAR3, and OVCAR5 cell lines, confirmed no significant cytotoxicity at 15 μM (see below **Figure, for Reviewer only**). While enzyme activity assays were not conducted in our study, Sheaff *et al.* reported a 45-60% inhibition of polymerases at a concentration of 5 μM ¹⁵. We have updated the Results with new references (#39 and #40) to justify the dose used in this study (**page 7**).

Cells were treated with aphidicolin (0-100 μM) for 48 hours. Cell growth was assessed using XTT assays ($n = 3$). Data are shown as mean \pm SD.

7a. Line 186. Here and in other places, the term “synergistic” is used but the data shown are mostly limited inhibition studies. For assessment of synergy, formal assessment (e.g., Combination Index) should be assessed and calculated.

7b. Line 202. As above, formal assessment of synergy should be done.

Response: Thank you for pointing these out. We agree with the Reviewer that a formal assessment of synergy by combination index would be difficult because we used a single dose for the

combination of aphidicolin and CHK1i or *POLE* silencing with CHK1i. Therefore, we have revised the sentences as follows for clarity (changes are shown in bold and *italics*).

Line 186 (now at **page 7**): “Notably, the combination of aphidicolin and CHK1i **synergistically significantly** inhibited cell growth in both cell lines (P<0.01, Fig. 4a)”.

Line 202 (now at **page 8**): “However, combining *POLE* silencing with CHK1i induced **synergistic cytotoxic effects** with ~50% growth inhibition in OVCAR3 cells (Fig. 4b), that augmented further with increasing CHK1i doses (Supplementary Fig. 3b)”.

8. Line 193. In this section, apparently siRNA pools were used. How was specificity of siRNA assessed? What was the extent of knockdown of the genes of interest?

Response: The efficacy of gene silencing was assessed through immunoblotting of POLA1 and POLE in cells transfected with siRNA pool for 48 hours (Fig. 4h). The bands corresponding to POLE and POLA1 in the respective lanes demonstrated clear evidence of >75% silencing (Fig. 4h). Below figure (**for Reviewer only**) shows silencing effects using densitometric analysis of the bands for each cell line from Fig. 4h.

The ImageStudio software was used to perform densitometric analysis. The protein levels of POLA1 and POLE were normalized with β -actin and expressed relative to the siRNA control.

The siRNA silencing efficiency is shown as % silencing.

9. Page 8. It would be important to show *in vivo* data; otherwise the data are not particularly convincing. Ideally this would be done in several models and also assess tissue-level effects.

Response: We acknowledge the importance of *in vivo* data for a comprehensive study. However, due to time constraints for the rebuttal, we respectfully propose conducting animal studies with POLE- or POLA1-depleted xenografts as a separate project for future research.

10. Page 7. There are platinum-sensitive and platinum-resistant versions of the OVCAR3 and OVCAR5 cells reported. What is the platinum IC₅₀ level of the cells that the authors are using?

Response: We conducted new XTT assays to assess the IC₅₀ values of cisplatin and carboplatin in OVCAR3 and OVCAR5 cells, employing BRCA1-null UWB1.289 as a platinum-sensitive control. The IC₅₀ values for OVCAR3 and OVCAR5 were 2- and 4.3-fold for cisplatin and 1.5- and 3.3-fold for carboplatin, respectively, compared to UWB1.289 (see below **Figures, for Reviewer only**).

Cells were treated with cisplatin (0-40 μM) or carboplatin (0-100 μM) for 48 hours. Cell growth was assessed using XTT assays (n = 3). IC₅₀ values were calculated using GraphPad Prism v9.

Data are shown as mean ± SD.

11. Page 11. Portions of the Discussion simply repeat the results. Would suggest re-doing the discussion to focus more on interpreting the results and presenting context rather than repeating results.

Response: Thank you. We have updated the Discussion section.

Reviewer #3 (Remarks to the Author): with expertise in ovarian cancer, therapy

The authors herein report the results of an important effort of translational research regarding the activity of prexasertib, a CHK1 inhibitor, in platinum-resistant high grade epithelial ovarian cancer (PROC).

It is noteworthy that this team has already published the results of the first proof of concept phase 2 study of prexasertib in *Lancet Oncol*, 2018. This preliminary assessment was conducted on 28 patients, of whom 24 were assessable. ORR was 33% in the treated population. In a much larger study published in *Gynecologic Oncology* (2022), Konstantinopoulos et al included 169 patients with PROC and found an ORR of 12.1 %.

In 2018, Eli Lilly decided to stop the development of prexasertib and the drug was sold to Acrivon in 2021.

In the present manuscript, the authors tried to find new biological predictive factors of prexasertib efficacy through multiple analyses of genomic and transcriptomic profiling, mutational signature, in vitro studies of cancer cell lines, and CTC / immune cells quantification and characterization.

The manuscript is well-written and the methodology of biological analyses is adequately described.

MAJOR COMMENTS

1. Despite multiple tests, it seems that the only pertinent finding in this study was the upregulation of genes involved in the DNA high-fidelity replication machinery, among which POLA1 could represent a potential target for future drugs, but which ones ?

To our knowledge, MIR002 could be a candidate [Dallavalle S, et al. Antitumor activity of novel POLA1-HDAC11 dual inhibitors. Eur J Med Chem. 2022, 228:113971.] but has never been tested in humans. This should be discussed by the authors.

Response: Thank you for your sharing insights. The Discussion part has been updated accordingly (page 12) and we agree that further research is needed for polymerase blockade as a therapeutic strategy in ovarian cancer.

2. This line of research seems to have been abandoned by Acrivon, which has adopted the OncoSignature companion test in its current basket study in gynecological tumors. The OncoSignature companion tests uses mass spectrometry, biased tumor model analyses, and quantitative multispectral in situ imaging of patient-derived xenografts to measure protein biomarkers and other biological mechanisms that are driving patients' tumors and can be interrogated by targeted drugs. The authors mentioned this approach in their conclusion but it should be further discussed.

Response: We acknowledge that OncoSignature companion test by Acrivon Therapeutics is one of the biomarkers of possibly predicting the CHK1i response^{16,17}. Also, we understand that the FDA recently granted a fast-track designation of ACR-368 (a.k.a., prexasertib) in OncoSignature-positive platinum-resistant recurrent ovarian cancer. GOG-3082 is a OncoSignature biomarker-based registration-intent phase II trial of ACR-368 (NCT055482962) that Dr. Lee serves as a Study Chair. We have updated Discussion accordingly (pages 13-14).

3a. One could be surprised by the number of patients who were finally not evaluable for response (10/49), which is unexpectedly high (13/169 in the pivotal study by Konstantinopoulos). Three patients withdraw their consent: was it due to toxicity ? One could imagine that patients with the poorest prognosis (sepsis, renal failure, fistula, etc.) could not complete the treatment. Thus, the ORR reported in evaluable but highly selected patients seems overestimated, which casts doubt on the translational analysis itself. Are some descriptive results available in the 10 "non assessable patients" and how do they compare to the overall results presented in the paper ?

Response: We appreciate the Reviewer's comments. As to the three patients who refused the further therapy after 1st or 2nd doses of prexasertib, the details are listed below;

1) One patient refused the therapy due to the development of bowel obstruction requiring surgery, **new Fig. 2b** is corrected to indicate this case as an intercurrent illness; bowel obstruction (n=1) instead of the patient refusal.

2) One patient wanted to try another clinical trial with a PARP inhibitor.

3) One patient developed non-neurocardiogenic syncope approximately ten days after the second dose of prexasertib. She was admitted for further evaluation then she refused the treatment.

This adverse event was determined not drug-related after consultation with the cardiology and neurology.

We acknowledge that Konstantinopoulos *et al.*⁴ reported 7.7% (13/169) of a withdrawal rate which seems to be a lower number compared to the current study. It is noteworthy that the study by Konstantinopoulos *et al.* included 4 cohorts with different eligibility criteria, with three of four cohorts including patients with BRCAwt, platinum-resistant disease. Of these three cohorts, only cohort 1 (platinum-resistant, BRCAwt patients with ≥ 3 prior lines of therapy; n=53) included a comparable patients population to our study with a median 4 prior lines of therapy. Of note, the median number of prior therapies for both cohort 2 (n=46, patients with up to 3 previous lines of therapy) and cohort 4 (n=29, patients with any number of prior lines) were two. It is unclear whether a majority of 13 patients who withdrew their consent belongs to this cohort 1 which can lead to a higher withdrawal rate than 7.7% if the comparison may be made between the cohort 1 of the study by Konstantinopoulos *et al.* and our study. Therefore, it would be hard to draw any conclusion between the two studies with limited information.

4. According to their statistical hypothesis, the authors should have included 71 patients in their trial. Only 39 were finally evaluable. Therefore, the study must be considered as inconclusive. It is difficult to interpret the results of ancillary studies in this context. Thresholds for significance are borderline and multiple tests do not allow to avoid results which would have been obtained by chance.

Response: Please see our response to Reviewer #1, Comment #1. Briefly, due to unforeseen circumstances, COVID-19, and Eli Lilly's termination of sponsorship for drug supplies (due to a licensing deal of prexasertib), cohorts A and B were prematurely closed before enrolling the planned numbers. We've submitted a clinical protocol amendment explaining the early closure (see NIH IRB approved- clinical protocol version date 11/13/2023 in the **Supplementary Information**, specifically the pages 2 and 67 of the protocol, cohorts 5 and 6 in the protocol = cohorts A and B in the manuscript).

Also please see our response to Reviewer #1, Comment #4. The Results and Discussion sections have been updated in consideration of the small sample size and possible false discovery (**pages 6-7 and page 12**). FDR q values for genes and pathways have been incorporated into **new Supplementary Tables 6-7**.

OTHER COMMENTS

5. Why were BRCA mutations excluded from the trial, since the BRCAm status has no particular prognostic influence in heavily pretreated PROC patients ? The sentence in the introduction l.60-61 should be modified : the unmet need is for both categories of patients.

Response: To clarify, the current manuscript is the first and final report of the new BRCAwt platinum-resistant HGSOc cohorts (cohorts 5 and 6 in the protocol [cohort A and B in the current manuscript]) of the NCI single-center phase 2 basket study of prexasertib. The clinical outcomes of prexasertib in *BRCA*-mutant HGSOc (cohort 3 in the protocol, see page 2 of the protocol Study Schema) was previously reported¹⁸ (# 22 in the manuscript).

As per the Reviewer's suggestion, we have revised the introduction for clarity (**page 3**, changes are shown in bold and *italics*): "~~Among these patients,~~***Platinum***-resistant recurrent HGSOc, ***particularly in BRCA wild-type (BRCAwt) cases (~75% of all HGSOcs)***^{3,4}, poses the greatest ~~treatment~~ challenge ***with limited treatment options***⁵, ~~Treatment options are limited in this setting,~~

highlighting an unmet need to develop novel therapeutic agents, particularly for the majority (~75%) of HGSOC patients who have no *BRCA1* or *BRCA2* mutation”.

6a. The distinction between primary and secondary platinum resistance is artificial and is not based of current evidence nor clinical guidelines.

Response: Thank you for your comments. We have no objection if the Reviewer strongly feels that this distinction should be removed. The reason why we made distinction for the primary vs secondary platinum resistance was because the follow up study of AURELIA trial clearly demonstrated that primary platinum resistance (<6 months from the last cycle of the first-line platinum-based chemotherapy) was associated with worse PFS and OS compared to secondary platinum resistance¹⁹.

6b. Fig 2.e shows nos difference in terms of response and PFS.

Response: To clarify, most of patients have their best response at the first CT scan evaluation (approximately two months after initiating the therapy), and this is consistent with our previous reports¹.

7. The authors should discuss the discrepancies between their study and the results presented in Gynecol Oncol, because it seems unlikely that they could be based on biological heterogeneity only.

Response: We agree with the Reviewer that it seems unlikely that this difference could be based on biological difference between the two study populations. Hence, as indicated by Konstantinopoulos *et al*⁴ Gynecol Oncol paper, more efforts for alternative biomarker approaches or combination therapies would be needed to extend the activity of prexasertib in patients with recurrent HGSOC.

Also, it is noteworthy that the median PFS appears to be similar between the two studies (5.6 months in Konstantinopoulos *et al.*'s study⁴ and 5 months in our current report) while ORR may

look different. This similarity underscores potential clinical benefits for a subset of patients in both studies, emphasizing the importance of exploring potential biomarkers for response and resistance to CHK1i treatment. We believe that conducting investigator-initiated comprehensive translational/correlative studies is important to advance our knowledge though they may be initially hypothesis-generating only.

Reviewer #4 (Remarks to the Author): with expertise in ovarian cancer, therapy

Synopsis of findings:

Clinical outcomes of the study are reported. They report an ORR of 30.7% in the RECIST-evaluable population with a median PFS of 5.8 month. This is a clinically meaningful result in this challenging to treat patient population.

No correlation between CCNE1 amplification and clinical response to prexasertib was identified. It is noted that only 2 patients had tumors demonstrating CCNE1 amplification, so it is perhaps more appropriate to say that this study could not meaningful confirm or refute the role for CCNE1 amplification as predictive biomarker.

Response: Thank you for your sharing enthusiasm. Of note, we corrected the median PFS as 5 months from 5.8 months because there was an error in one patient's PFS data. We believe this is still clinically meaningful PFS given the median PFS in this heavily pretreated population is usually 3-4 months to single agent chemotherapy^{11,12}. Although Mirvetuximab Soravtansine (MS) yielded a median PFS of 5.62 months and 42.3% ORR, this registration trial limited the previous lines of therapy to up to three and only the population with "high" FR α tumor expression was eligible¹². Therefore, we believe our data are still meaningful as more novel therapeutic options are needed after bevacizumab and MS-exposure in heavily pretreated platinum-resistant HGSOc.

We have revised this sentence (**page 11**) as follows (changes are shown in bold and *italics*):
“However, we could not confirm or refute the role of CCNE1 alone is not enough to assess the dynamic nature of RS, a single amplification as predictive biomarker will be unlikely for

~~predicting CHK1i response/resistance in BRCAwt HGSOC because only two patients had tumors demonstrating CCNE1 amplification”.~~

The authors present data from NGS and whole exome sequencing. No significant trends or predictors are identified.

The authors conducted transcriptomic analysis and identified a significant mRNA overexpression of POLA1 and POLE in the patients without clinical benefit, as well as overexpression of MCM7 and GINS3. The authors hypothesize that upregulation of high-fidelity polymerases (POLA1, POLE) may be critical for DNA synthesis in BRCAwt platinum-resistant HGSOC.

They subsequently tested this hypothesis in cell lines (OVCAR3 and OVCAR5, platinum-resistant BRCAwt HGSOC cell lines) using a pan-inhibitor of B-family polymerases. The combination of a pan-inhibitor of B-family polymerases with prexasertib demonstrated synergistic cell growth inhibition.

siRNA targeting POLA1 and POLE were used to identify the respective role of each. POLA1 silencing resulted in inhibition of growth in both OVCAR3 and OVCAR5 cell lines.

They subsequently evaluated mutational signatures and found significantly higher SBS7d and ID13 mutational signatures in the CB group compared to the NCB group. They propose that these cells benefiting from CHK1i may be more dependent on the error-prone translesion synthesis pathway. Notably, transcriptomic analysis did not reveal upregulation of translesion synthesis polymerase genes.

They examined changes in circulating tumor cells and systemic immune cells. No definitive conclusions were drawn from CTC analysis. An increase in myeloid-derived suppressor cells and classical monocytes was observed following exposure to prexasertib in the non-clinical benefit group.

Significance to the field:

Therapies are greatly needed for patients with platinum-resistant recurrent, HGSOC. The results of this phase 2 trial represent a meaningful clinical result in this population.

As the authors point out on multiple occasions, there is considerable molecular and genetic heterogeneity in the HGSOC population. This problem is only compounded by the complex and dynamic nature of replication stress response. This work is an ambitious undertaking to identify potential predictive biomarkers to CHK1 inhibition.

Several novel findings are reported, including the potential role of POLA1 and POLE in the response to CHK1 inhibition.

Specific comments to authors:

1. This study is limited to patients with BRCA wild type disease. The rationale for this limitation is presented in the discussion but could be more clearly presented in the introduction.

Response: We have revised the introduction (**page 3**) as follows (changes are shown in bold and *italics*): ~~”Among these patients,~~***Platinum***-resistant recurrent HGSOC,***particularly in BRCA wild-type (BRCAwt) cases (~75% of all HGSOCs)***^{3,4}, poses the greatest ~~treatment~~ challenge ***with limited treatment options***⁵, ~~Treatment options are limited in this setting,~~ highlighting an unmet need to develop novel therapeutic agents,~~particularly for the majority (~75%) of HGSOC patients who have no BRCA1 or BRCA2 mutation”.~~

2. Line 213 – note typo in the spelling of prexasertib

Response: Corrected.

3. By performing exploratory analyses in cell lines, albeit 2 different lines, many of the findings presented here may not be reproducible in a broader HGSOC population. Further research to address this consideration would be warranted, and the need for further investigation into these findings is generally proposed. I would recommend making more explicit the limitations in the application of these findings to the broader landscape of HGSOC.

Response: We have updated the Discussion (**page 13**) as follows: “Lastly, we acknowledge the potential limitations of exploratory findings from analyses in two cell lines to a broader HGSOC population, requiring further studies in various preclinical and clinical HGSOC models.”

4. Methodology:

The methodology appears to be rigorous, both in the execution of the clinical trial reported and in the subsequent hypothesis-generating preclinical studies.

Response: Thank you.

References

- 1 Lee, J. M. *et al.* Prexasertib, a cell cycle checkpoint kinase 1 and 2 inhibitor, in BRCA wild-type recurrent high-grade serous ovarian cancer: a first-in-class proof-of-concept phase 2 study. *Lancet Oncol* **19**, 207-215 (2018). [https://doi.org/10.1016/S1470-2045\(18\)30009-3](https://doi.org/10.1016/S1470-2045(18)30009-3)
- 2 Wang, T. *et al.* Effect of Apatinib Plus Pegylated Liposomal Doxorubicin vs Pegylated Liposomal Doxorubicin Alone on Platinum-Resistant Recurrent Ovarian Cancer: The APPROVE Randomized Clinical Trial. *JAMA Oncol* **8**, 1169-1176 (2022). <https://doi.org/10.1001/jamaoncol.2022.2253>
- 3 Matulonis, U. A. *et al.* Efficacy and Safety of Mirvetuximab Soravtansine in Patients With Platinum-Resistant Ovarian Cancer With High Folate Receptor Alpha Expression: Results From the SORAYA Study. *J Clin Oncol* **41**, 2436-2445 (2023). <https://doi.org/10.1200/JCO.22.01900>
- 4 Konstantinopoulos, P. A. *et al.* A Phase 2 study of prexasertib (LY2606368) in platinum resistant or refractory recurrent ovarian cancer. *Gynecol Oncol* **167**, 213-225 (2022). <https://doi.org/10.1016/j.ygyno.2022.09.019>
- 5 Poplawski, A. & Binder, H. Feasibility of sample size calculation for RNA-seq studies. *Brief Bioinform* **19**, 713-720 (2018). <https://doi.org/10.1093/bib/bbw144>
- 6 Subramanian, A. *et al.* Gene set enrichment analysis: a knowledge-based approach for interpreting genome-wide expression profiles. *Proc Natl Acad Sci U S A* **102**, 15545-15550 (2005). <https://doi.org/10.1073/pnas.0506580102>
- 7 Altorki, N. K. *et al.* Neoadjuvant durvalumab plus radiation versus durvalumab alone in stages I-III non-small cell lung cancer: survival outcomes and molecular correlates of a randomized phase II trial. *Nat Commun* **14**, 8435 (2023). <https://doi.org/10.1038/s41467-023-44195-x>
- 8 Zhou, Y., Lei, D., Hu, G. & Luo, F. A Cell Cycle-Related 13-mRNA Signature to Predict Prognosis in Hepatocellular Carcinoma. *Front Oncol* **12**, 760190 (2022). <https://doi.org/10.3389/fonc.2022.760190>
- 9 Cancer Genome Atlas Research, N. Integrated genomic analyses of ovarian carcinoma. *Nature* **474**, 609-615 (2011). <https://doi.org/10.1038/nature10166>
- 10 Pennington, K. P. *et al.* Germline and somatic mutations in homologous recombination genes predict platinum response and survival in ovarian, fallopian tube, and peritoneal carcinomas. *Clin Cancer Res* **20**, 764-775 (2014). <https://doi.org/10.1158/1078-0432.CCR-13-2287>
- 11 Pujade-Lauraine, E. & Combe, P. Recurrent ovarian cancer. *Ann Oncol* **27 Suppl 1**, i63-i65 (2016). <https://doi.org/10.1093/annonc/mdw079>
- 12 Moore, K. N. *et al.* Mirvetuximab Soravtansine in FRalpha-Positive, Platinum-Resistant Ovarian Cancer. *N Engl J Med* **389**, 2162-2174 (2023). <https://doi.org/10.1056/NEJMoa2309169>
- 13 Erba, E., Sen, S., Lorico, A. & D'Incalci, M. Potentiation of etoposide cytotoxicity against a human ovarian cancer cell line by pretreatment with non-toxic concentrations of methotrexate or aphidicolin. *Eur J Cancer* **28**, 66-71 (1992). [https://doi.org/10.1016/0959-8049\(92\)90387-h](https://doi.org/10.1016/0959-8049(92)90387-h)
- 14 Sargent, J. M., Elgie, A. W., Williamson, C. J. & Taylor, C. G. Aphidicolin markedly increases the platinum sensitivity of cells from primary ovarian tumours. *Br J Cancer* **74**, 1730-1733 (1996). <https://doi.org/10.1038/bjc.1996.622>

- 15 Sheaff, R., Ilsley, D. & Kuchta, R. Mechanism of DNA polymerase alpha inhibition by aphidicolin. *Biochemistry* **30**, 8590-8597 (1991). <https://doi.org/10.1021/bi00099a014>
- 16 Ayesha Murshid, C. N., Kailash Singh, Sibgat Choudhury, Jayakumar Nair, Lei Shi, James Duniak, Jung-Min Lee, Kristina Masson, Michail Shipitsin, Peter Blume-Jensen. in *AACR-NCI-EORTC International Conference on Molecular Targets and Cancer Therapeutics* (Boston, Massachusetts, 2023).
- 17 Caroline Wigerup, M. S., Ayesha Murshid, Lei Shi, Magnus E. Jakobsson, Shahrzad Rafiei, Dorte Bekker-Jensen, Sibgat Choudhury, Jim Duniak, Joon Jung, David Proia, Jesper V. Olsen, Kristina Masson, Peter Blume-Jensen. in *AACR-NCI-EORTC International Conference on Molecular Targets and Cancer Therapeutics* (Boston, Massachusetts, 2023).
- 18 Gupta, N. *et al.* BLM overexpression as a predictive biomarker for CHK1 inhibitor response in PARP inhibitor-resistant BRCA-mutant ovarian cancer. *Sci Transl Med* **15**, eadd7872 (2023). <https://doi.org/10.1126/scitranslmed.add7872>
- 19 Trillsch, F. *et al.* Prognostic and predictive effects of primary versus secondary platinum resistance for bevacizumab treatment for platinum-resistant ovarian cancer in the AURELIA trial. *Ann Oncol* **27**, 1733-1739 (2016). <https://doi.org/10.1093/annonc/mdw236>

REVIEWERS' COMMENTS

Reviewer #1 (Remarks to the Author):

I have no further statistical concern. One minor suggestion is to consider including the rationale of comparability of the two cohorts in the supplementary material. The rebuttal letter has detailed analysis for testing statistical equivalence of Cohort A and B.

Reviewer #2 (Remarks to the Author):

The revised manuscript has incrementally addressed some of the comments:

- 1) Prior comment 4. The main question here was why use “clinical benefit rate”? It is a relatively soft endpoint and not usable for drug approval. ORR would be a better distinction.
- 2) Prior comment 8. The response doesn't really address specificity of the siRNA sequences – they have simply shown the extent of silencing.
- 3) Prior comment 9. Without in vivo data, the findings are not particularly convincing.

Reviewer #4 (Remarks to the Author):

I thank the authors for submitting this revised version of the manuscript. Each of my comments has been systematically and adequately addressed.

Reviewer #1:

I have no further statistical concern. One minor suggestion is to consider including the rationale of comparability of the two cohorts in the supplementary material. The rebuttal letter has detailed analysis for testing statistical equivalence of Cohort A and B.

Response: We have included the rationale of comparability of the two cohorts in Supplementary Data 1 and Methods (page 20).

Reviewer #2:

The revised manuscript has incrementally addressed some of the comments:

1) Prior comment 4. The main question here was why use “clinical benefit rate”? It is a relatively soft endpoint and not usable for drug approval. ORR would be a better distinction.

Response: We appreciate the Reviewer’s comment on ORR as a better distinction for the drug approval although PFS and OS are preferred endpoints for oncology drug approval¹. Currently, prexasertib (a.k.a. ACR-368) is being investigated for drug approval in platinum-resistant ovarian cancer and ORR is a primary endpoint in this registration intent phase 2 study (NCT05548296) which has a larger sample size than our NCI phase 2 pilot study.

The aim of the NCI phase 2 pilot study was to identify early clinical activity of the drug in platinum-resistant ovarian cancer and to conduct comprehensive translational studies using before and on treatment fresh biopsy and blood samples. Given the small sample size with tissue biopsies suitable for transcriptomic analyses (n=17 including 11 patients with stable diseases and 6 patients with partial responses), we elected to use a PFS over ORR to have a reasonable number for analysis and to better split our population with RNAseq data into two parts. As shown on Fig. 2c, our data indicate that “clinical benefit rate” based on PFS better distinguishes the two populations of “responders” and “non responders”. There are more patients with a sustained stable disease (PFS \geq 6 months, 10 patients), while there is one patient with partial response lasting < 6 months. Also, it is worth noting that a stable disease lasting longer than 6 months is clinically meaningful in the platinum-resistant setting because median PFS is usually 3-4 months to single agent chemotherapy^{2,3}.

2) Prior comment 8. The response doesn't really address specificity of the siRNA sequences – they have simply shown the extent of silencing.

Response: The ON-TARGETplus Smartpool siRNA tailored for *POLE* and *POLA1* is a mixture of four carefully designed siRNAs. These sequences, detailed in Supplementary Data 8, are designed using proprietary algorithms and synthesized with patented modifications to ensure increased specificity for the target mRNA.

To assess siRNA specificity, we performed western blot analyses in cells transfected with siRNAs against *POLE* and *POLA1*. Our results showed no change in *POLE* protein levels upon silencing of *POLA1* and similarly no change in *POLA1* proteins upon silencing of *POLE* (Fig. 4h). Furthermore, no discernible off-target effects were observed on the loading marker actin within the same samples, further confirming the specificity of the siRNAs used in this study.

3) Prior comment 9. Without in vivo data, the findings are not particularly convincing.

Response: We appreciate your feedback on the importance of *in vivo* data. Unfortunately, there are no clinically developed *POLE* or *POLA1* inhibitors available for animal studies. While promising candidates exist, such as *POLA1*-HDAC1 dual inhibitor MIR002⁴ or *POLA1* inhibitor ST1926⁵, none of them has been tested in humans.

We acknowledge the limitations of transient gene knockdown in cell line data, thus respectfully propose establishing stable *POLE*- or *POLA1*-knockout ovarian cancer cell lines and specific inhibitors as a separate project for future studies. We have addressed this limitation in the Discussion section (page 13).

Reviewer #4:

I thank the authors for submitting this revised version of the manuscript. Each of my comments has been systematically and adequately addressed.

Response: Thank you.

References

- 1 Saad, E. D. & Buyse, M. Statistical controversies in clinical research: end points other than overall survival are vital for regulatory approval of anticancer agents. *Ann Oncol* **27**, 373-378 (2016). <https://doi.org/10.1093/annonc/mdv562>
- 2 Pujade-Lauraine, E. & Combe, P. Recurrent ovarian cancer. *Ann Oncol* **27 Suppl 1**, i63-i65 (2016). <https://doi.org/10.1093/annonc/mdw079>
- 3 Moore, K. N. *et al.* Mirvetuximab Soravtansine in FRalpha-Positive, Platinum-Resistant Ovarian Cancer. *N Engl J Med* **389**, 2162-2174 (2023). <https://doi.org/10.1056/NEJMoa2309169>
- 4 Dallavalle, S. *et al.* Antitumor activity of novel POLA1-HDAC11 dual inhibitors. *Eur J Med Chem* **228**, 113971 (2022). <https://doi.org/10.1016/j.ejmech.2021.113971>
- 5 Zuco, V., Benedetti, V., De Cesare, M. & Zunino, F. Sensitization of ovarian carcinoma cells to the atypical retinoid ST1926 by the histone deacetylase inhibitor, RC307: enhanced DNA damage response. *Int J Cancer* **126**, 1246-1255 (2010). <https://doi.org/10.1002/ijc.24819>